# Structure and dynamics of 2x(CENP-A/H4)$_2$ octasome reveal a possible intermediate in centromeric chromatin

Ahmad Ali-Ahmad[1] , Mira Mors[2] , Manuel Carrer[2], Xinmeng Li[2], Silvija Bilokapić[3], Mario Halić[3], Michele Cascella[2,4] , Nikolina Sekulić[1,2,5]

**The centromere is a part of the chromosome that is essential for the even segregation of duplicated chromosomes during cell division. It is epigenetically defined by the presence of the histone H3 variant CENP-A. CENP-A associates specifically with a group of 16 proteins that form the constitutive centromere-associated network (CCAN) of proteins. In mitosis, the kinetochore forms on the CCAN to connect the duplicated chromosomes to the microtubules protruding from the cell poles. Previous studies have shown that CENP-A replaces H3 in nucleosomes, and recently, the structures of CENP-A–containing nucleosomes in complex with CCANs have been revealed, but they show only a limited interaction between CCANs and CENP-A. Here, we report the cryo-EM structure of 2x(CENP-A/H4)$_2$ octasomes assembled on DNA in the absence of H2A/H2B histone dimer and speculate how (CENP-A/H4)$_2$ tetrasomes might serve as a platform for CCAN organization.**

## Introduction

The centromere plays a crucial role in the correct distribution of genetic material during cell division by ensuring that the duplicated chromosomes are divided evenly between two new cells (1, 2, 3, 4, 5). The centromere is epigenetically defined by the presence of an H3 histone variant called CENP-A (reviewed in reference 6). CENP-A is instrumental in the organization of the constitutive centromere-associated network (CCAN) of 16 proteins. Despite major advances in obtaining high-resolution structures of CENP-A nucleosomes (7, 8) and CCANs (9, 10) and even cryo-EM structures of human and yeast CENP-A nucleosomes in complex with CCANs (11, 12, 13), it is still not clear which CENP-A features control CCAN organization in CENP-A–enriched parts of chromatin. Furthermore, it has been proposed that CCAN structure and the nature of its

interaction with CENP-A change dynamically during the cell cycle (1, 14, 15, 16).

There are several indications that only two CCAN components, CENP-C and CENP-N, can interact directly with CENP-A and that the remaining proteins form subcomplexes on this structure ((17) also reviewed in reference 18). The two segments on the human CENP-C (CENP-C$^{426–537}$ and CENP-C$^{737-359}$) recognize and bind the CENP-A nucleosome by interlocking with its C-terminal tail, which is exposed on the surface of the nucleosome and the acidic patch on histone H2A that is also exposed on the surface of the nucleosome. The interaction between CENP-C and CENP-A nucleosome has been visualized in several studies, including the CENP-A/CCAN structures of yeast and human complexes (8, 11, 12, 13, 16, 19, 20). On the other hand, CENP-N was found to interact with the L1 loop of CENP-A, which has an RG insertion (compared with the H3 histone) (21). The RG loop is also exposed on the surface of the CENP-A nucleosome, and several cryo-EM structures have captured the interaction of the N-terminal portion of CENP-N with the CENP-A$^{RG-loop}$ and DNA on the CENP-A nucleosome (16, 22, 23, 24). However, when full-length CENP-N is presented together with the binding partner CENP-L in the context of CCANs, the CENP-N/L complex does not bind the CENP-A$^{RG-loop}$ on CENP-A nucleosomes, but rather binds only DNA protruding from the nucleosome and far away from the histone core (11, 12, 13). This leaves open the question of the context in which CENP-A and CENP-N interact directly via the CENP-A$^{RG-loop}$ and whether it is possible that either CENP-A or CCAN has an alternative organization in chromatin that allows direct interaction between CENP-A and CENP-N during the cell cycle.

Previous high-resolution structures of CENP-A show either (CENP-A/H4)$_2$ tetramers without DNA (25) or nucleosomes (multiple structures, in isolation or in complex with other proteins), suggesting that these are the most stable complexes of CENP-A. Several proposals for CENP-A–containing particles have been made in the past (reviewed in reference 26) including a CENP-A "hemisome" structure in which only one copy of each histone (CENP-A, H4, H2A, and H2B) is present.

[1]Norwegian Centre for Molecular Biosciences and Medicine (NCMBM), Nordic EMBL Partnership, Faculty of Medicine, University of Oslo, Oslo, Norway    [2]Department of Chemistry, University of Oslo, Oslo, Norway    [3]Department of Structural Biology, St. Jude Children's Research Hospital, Memphis, TN, USA    [4]Hylleraas Centre for Quantum Molecular Sciences, University of Oslo, Oslo, Norway    [5]Department of Molecular Medicine, Institute of Basic Medical Sciences, Faculty of Medicine, University of Oslo, Oslo, Norway

Correspondence: a.a.ahmad@ncmbm.uio.no; nikolina.sekulic@medisin.uio.no
Mira Mors's present address is Institute for Theoretical Physics IV, University of Stuttgart, Stuttgart, Germany

However, this particle has not been stabilized in vitro for high-resolution studies, nor has it been observed in cells ([27]). To reconcile the cryo-EM structure of CCANs with previous observations of CENP-A/CENP-N interaction, Musacchio's lab proposed a "hemisome" in which CENP-T/CENP-W replaces H2A/H2B, but the high-resolution structure of the particle was never reported ([9]).

Here, we show that (CENP-A/H4)$_2$ can form an octasome structure on DNA in vitro in the absence of H2A/H2B histones. We report two different conformations captured by cryo-EM that we call open and closed. The molecular dynamics (MD) analysis finds that these structures are stable but highly dynamic, and we provide evidence that 2x(CENP-A/H4)$_2$ octasomes can form a unique chromatin structure in vitro. We also show that 2x(CENP-A/H4)$_2$ octasome can bind CENP-C albeit with lower affinity than the CENP-A nucleosome and that all CENP-A–containing particles can bind CENP-N/L complex. We discuss the implications of our findings on the possible interactions between CENP-A and CCANs.

## Results

### Two (CENP-A/H4)$_2$ tetramers assemble into 2x(CENP-A/H4)$_2$ octasome on DNA

We incubated (CENP-A/H4)$_2$ tetramers or (H3/H4)$_2$ tetramers with 147-bp DNA in a 2:1 ratio in the absence of the H2A/H2B dimer (Figs 1A and S3A), and we observed formation of two types of particles with different mobility on the native gel (Fig S1A). We notice that by increasing the amount of (CENP-A/H4)$_2$ relative to DNA, we enrich the faster traveling particle, so we concluded that the particle corresponding to the slower band on gel is a tetrasome, whereas the one that travels faster and almost the same as the CENP-A nucleosome is the 2x(CENP-A/H4)$_2$ octasome (Fig S1A and C). This is similar to observation of 2x(H3/H4)$_2$ octasomes by reference [28] and recently reported 2x(CENP-A/H4)$_2$ octasomes by the same group ([29]). Indeed, we could assemble octasomes with both (H3/H4)$_2$ and (CENP-A/H4)$_2$ tetramers on 601 positioning and on natural centromeric alpha-satellite DNA, albeit 2x(CENP-A/H4)$_2$ octasomes consistently assemble with lower efficiency (Fig S1B). Next, we wondered whether octasomes would protect wrapped DNA in the same way as nucleosomes, so we performed MNase digestion of assembled particles (Fig S1E). Not surprisingly, nucleosomes wrap and protect more DNA than octasomes, but octasomes nevertheless show some DNA protection that is less pronounced for 2x(CENP-A/H4)$_2$ octasomes compared with their H3-containing counterparts. To better understand these structures, we obtained the cryo-EM maps of 2x(CENP-A/H4)$_2$ octasomes to 4.1 Å (Figs 1B and S2; Table S1) and 2x(H3/H4)$_2$ octasomes to 3.75 Å resolution (Fig S3B; Table S1).

### 2x(CENP-A/H4)$_2$ octasome forms a flexible nucleosome-like particle

2x(CENP-A/H4)$_2$ octasomes have the same overall structure as the 2x(H3/H4)$_2$ octasome ([28]) and the recently published 2x(CENP-A/H4)$_2$ octasome ([29]). To be able to compare the structures directly, we obtained the cryo-EM structure of both 2x(H3/H4)$_2$ and 2x(CENP-A/H4)$_2$ octasomes in the same conditions. Analysis of cryo-EM data indicates that octasomes are dynamic structures that are less stable during freezing, which is also in agreement with higher susceptibility to MNase digestion (Fig S1E). For each of the octasome particles, we refined two different classes, open and closed. The open class corresponds to particles that exhibit high degree of DNA unwrapping, whereas the closed class exhibits the most pronounced DNA wrapping (Figs 1B, S2B, and S3D).

The 2x(CENP-A/H4)$_2$ octasome has the H4/H4 four-helix bundle at the dyad (whereas the CENP-A nucleosome has CENP-A/CENP-A in this position), and two CENP-A/CENP-A four-helix bundles are on the superhelical location 3 (SHL3) and -3 (SHL-3) (Fig 2A–C). In both CENP-A and H3 nucleosomes, this position has H4/H2B four-helix bundle (Fig 2A–C). Also immediately noticeable is a greater separation of DNA gyres on the opposite side of the dyad (Fig 2D). The detailed structural analysis reveals several differences between nucleosomes and octasomes that result in increased gapping of the latter.

The octasomes accommodate H4/H4 four-helix bundle at the dyad, whereas nucleosomes have CENP-A/CENP-A or H3/H3 in this position. The hydrogen bonding network at the H3/H3 or CENP-A/CENP-A four-helix bundle is robust and well preserved with strong hydrogen bonds ranging in size between 2.6 and 2.9 Å. The arginine and asparagine on one chain and histidine on the opposite chain are present and available for hydrogen bonding in both H3 and CENP-A interfaces, and they form a strong RDH network: H3$^{Arg116}$, H3$^{Asp123}$, and H3$^{His113}$ in canonical and CENP-A$^{Arg118}$, CENP-A$^{Asp125}$, and CENP-A$^{His115}$ in CENP-A nucleosomes. Interestingly, similar RDH network exists at the H4-H4 interface in octasomes (residues H4$^{Arg78}$, H4$^{Asp85}$, and H4$^{His75}$), but here the distances between residues and their orientations are distorted (Fig 2B). In both nucleosome and octasome, the rest of the interface at the dyad is mediated by hydrophobic residues (in nucleosome, there are three leucine and one isoleucine residues, whereas in octasome, there are mainly aromatic residues [H4$^{Tyr72}$, H4$^{His75}$, and H4$^{Tyr88}$]). Because of loose hydrogen bonding, most of the interaction in H4/H4 interface at the dyad of octasomes is hydrophobic. This allows sliding at the four-helix bundle resulting in drastically different angles (~40° rotation) between a2 helices (Fig S4A) in octasomes compared with CENP-A nucleosomes.

We analyzed the differences at the SHL3 and SHL-3 four-helix bundle interfaces (Fig 2C). Here, octasomes have CENP-A/CENP-A or H3/H3 four-helix bundles with weak RDH network, whereas nucleosomes have heterologous H4/H2B four-helix bundles mediated mainly by hydrophobic interactions and stacking of aromatic residues (H4$^{Y72}$, H4$^{Y88}$, H2B$^{Y83}$, and H2B$^{L80}$). Thus, the interface of H4/H2B four-helix bundles exhibits fewer electrostatic interactions than the interfaces of H3/H3 or CENP-A/CENP-A at SHL3/SHL-3 in octasomes, suggesting that the compaction and stability of the nucleosome are not solely determined by interactions in the four-helix bundles. In fact, the distance between DNA gyres (gapping) is much tighter in nucleosomes than in octasomes (Fig 2D). The low gapping of the nucleosome is most likely caused by a beta-sheet within the histone core (highlighted in Fig 2E), which is formed between the beta-strand of H4 and the beta-

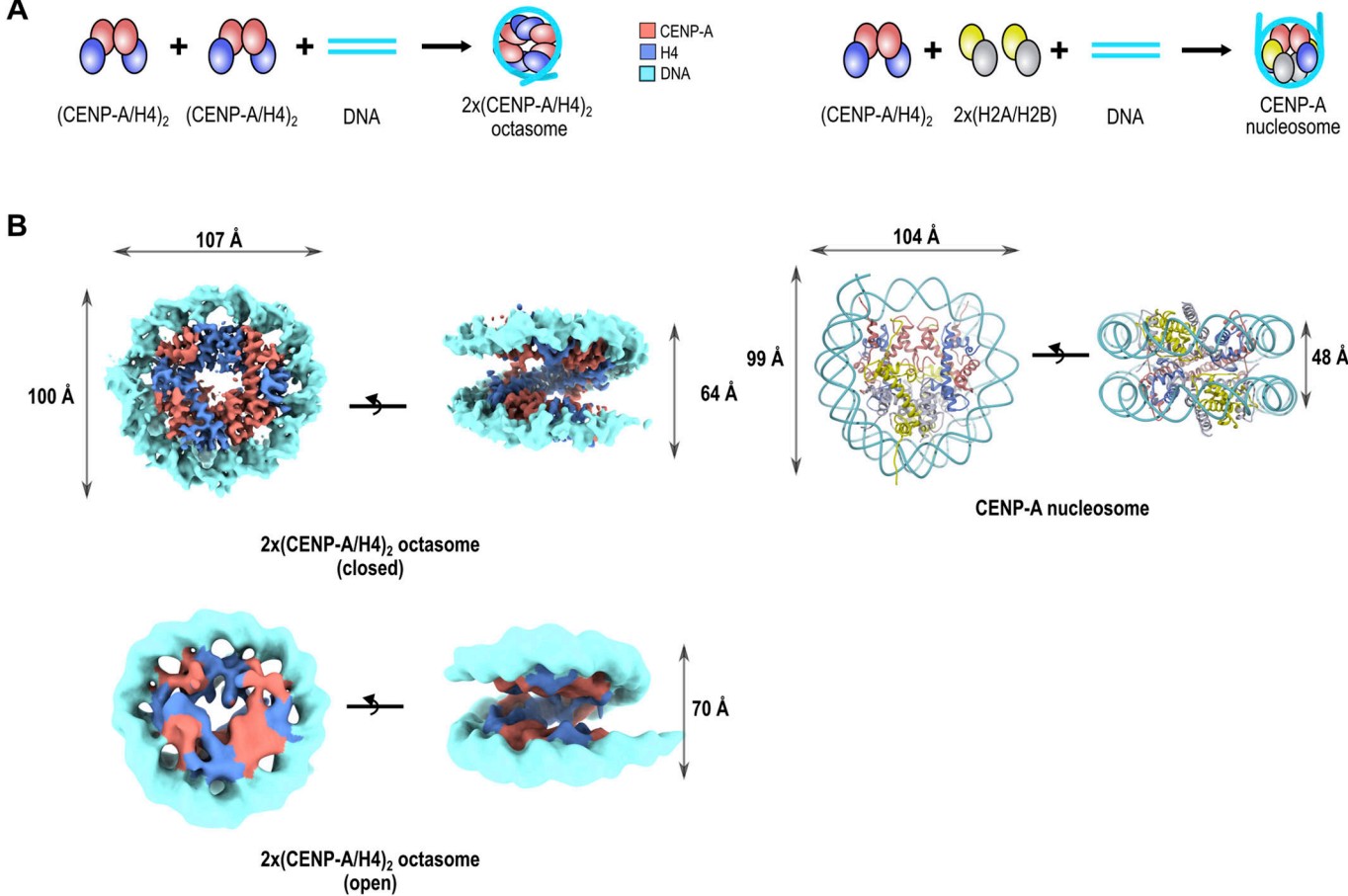

**Figure 1. Two (CENP-A/H4)₂ tetramers can assemble on DNA to form 2x(CENP-A/H4)₂ octasomes.**
**(A)** Schematic representation of the assembly of 2x(CENP-A/H4)$_2$ octasomes (left) and CENP-A nucleosomes (right) for comparison. CENP-A is red; H4 is blue; H2A is yellow; H2B is gray; and 145-bp DNA is cyan. **(B)** Cryo-EM map of the 2x(CENP-A/H4)$_2$ octasome assembled on 147-bp α-satellite DNA (left) in open and closed conformation showing the different distances between the DNA gyres (gapping) compared with the model CENP-A nucleosome (PDB 6SE0) on the right. Histones and DNA are colored as in (A). Note that the 2x(CENP-A/H4)$_2$ octasome four-helix bundle at the dyad is formed by H4-H4 interactions.

strand of H2A. Here, the H4 residues (H4$^{96-98}$) following the alpha-3 helix of H4 form a beta-strand with the H2A residues (H2A$^{100-102}$) following the a3 helix of H2A, whereas the remaining C-terminal residues of H2A are tightly stacked along the nucleosome face. The short beta-sheet between H4 and H2A holds the nucleosome together forming a stable particle. Interestingly, the same beta-sheet is reinforced (less solvent-exposed) in CENP-A nucleosomes as an allosteric effect of CENP-C binding (30). This "stitching" interaction within the nucleosome is not present in the octasomes. In addition, both H3- and CENP-A–containing octasomes lack the C-terminal tail of H2A, which could further stick the two halves of octasomes together. In contrast, both H3 and CENP-A have a long N-tail preceding the α1 helix. This N terminus contains a bulky, partially structured αN helix (not visible in our cryo-EM data; highlighted in Fig 2E) facing the interior of a clamshell-like structure that pushes away two halves of the particle.

The presence of bulky alpha-N-helices of CENP-A that are opposing the dyad generates tension and leads to a rotation of the H4 four-helix bundle to a final 46° divergence with respect to the CENP-A four-helix bundles at the nucleosome dyad (Fig S4A). These changes contribute to a more "breathable" structure with

increased flexibility and reduced DNA wrapping capacity, as reflected by more open and twisted DNA gyres on the opposite side of the dyad compared with a canonical nucleosome (Fig 2E). This is all consistent with the observation of open and closed conformations for octasomes, which are only two of many possible arrangements that we were able to refine from our cryo-EM data.

Thanks to this high flexibility, the octasomes can propagate on DNA to make longer "slinky-like" structures similarly as it was already reported for archaeal histones (31, 32). Indeed, by assembling (CENP-A/H4)$_2$ on longer DNA, we have observed by cryo-EM that (CENP-A/H4)$_2$ tetramers can form "slinkies" (Fig S5C; Table S2). Because octasomes are not so compact, the DNA at the entry/exit sites is further away from each other, so DNA packed with octasome has distinguishingly different architecture than chromatin packed in nucleosomes (Fig S5D).

### 2x(CENP-A/H4)₂ octasomes are stable but dynamic

To further investigate whether octasomes are stable, we performed molecular dynamics (MD) studies on CENP-A– and H3-containing octasomes (complete models with full-length histones

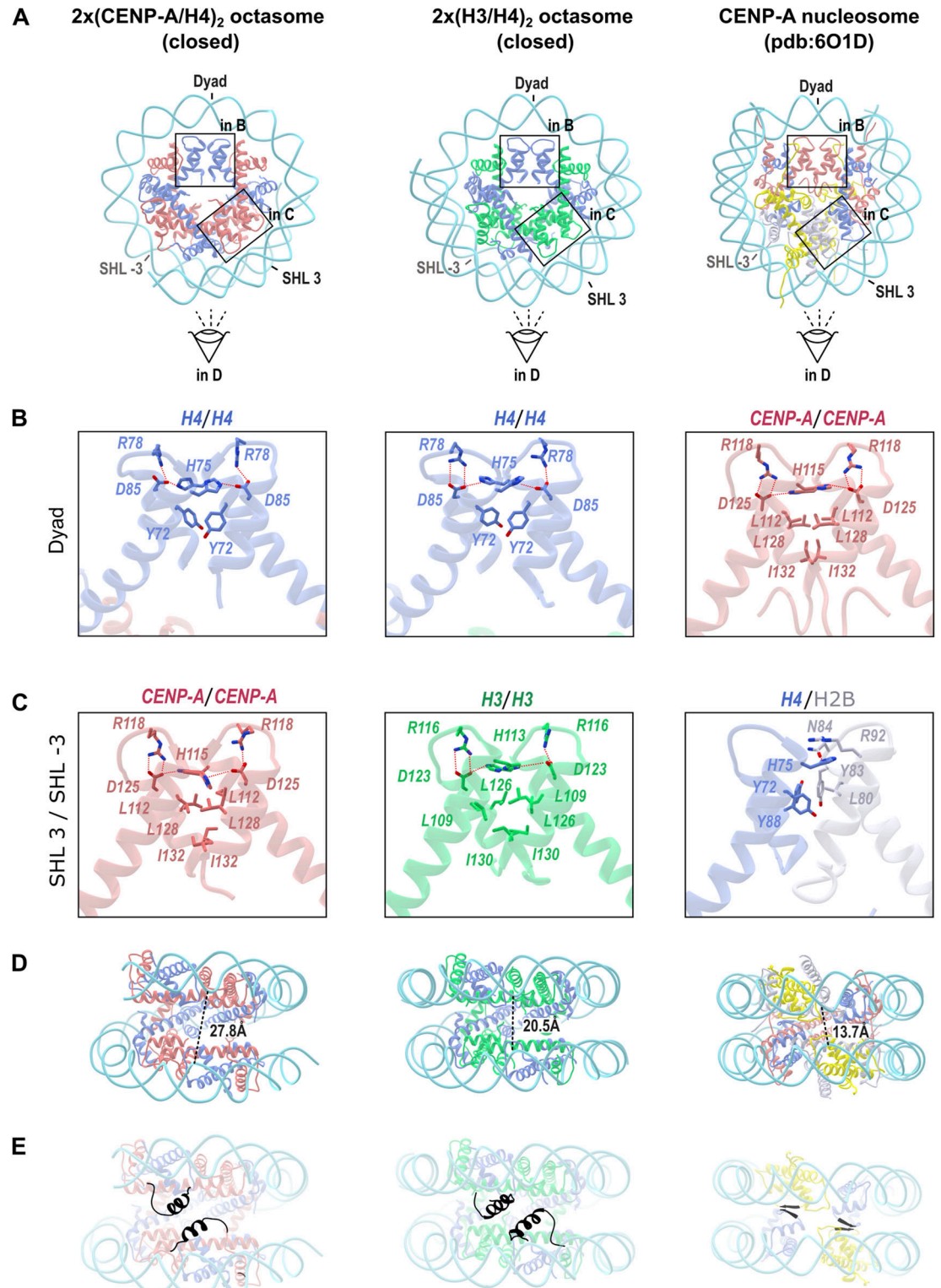

**Figure 2. Octasomes have a rotated H4/H4 four-helix bundle at the dyad and more separated DNA gyres.**

**(A)** Ribbon representations of 2x(CENP-A/H4)$_2$ octasome in closed conformation (left), 2x(H3/H4)$_2$ octasome in closed conformation (middle) and CENP-A nucleosomes (PDB 6O1D) (right). CENP-A is red; H3 is green; H4 is blue; H2A is yellow; H2B is gray; and 145-bp DNA is cyan. **(B, C)** Four-helix bundles at the dyad and at SHL3 are marked with rectangles and shown enlarged in (B, C), respectively. **(D, E)** Viewing direction for the side projection in panels (D, E) is indicated by an eye symbol. **(B)** Enlarged view of the four-helix bundle at the dyad for the 2x(CENP-A/H4)$_2$ octasome (left), the 2x(H3/H4)$_2$ octasome (middle), and the CENP-A nucleosome (right). The key residues responsible for the interaction at the four-helix bundles are shown in sticks. **(C)** Enlarged view of the four-helix bundle at SHL3 and SHL-3 for the 2x(CENP-A/H4)$_2$ octasome (left), the 2x(H3/H4)$_2$ octasome (middle), and the CENP-A nucleosome (right). **(D)** Side view of the particles from panel A (rotated clockwise by 90° around the horizontal axis) to illustrate the separation of the DNA gyres. Measurements were made from base pairs labeled as −36 and 40 in chain J of DNA for

and DNA) and nucleosomes (PDB 6O1D for the CENP-A nucleosome and PDB 3LZ0 for H3 nucleosome) (Fig 3; Table S2). MD studies indicate that the octasomes are structurally stable but at the same time exhibit greater flexibility compared with nucleosome particles (Figs 3 and S6). The topological distribution of the most mobile regions in both the core histones and DNA is illustrated by the heat map, in which the residues that make up the protein/DNA complexes are colored according to their calculated B-factor (Fig 3A). The root mean square fluctuations (RMSF) for the histone fold domains (HFD) of octasomes are generally higher than those of nucleosomes (Figs 3A and S6A–D). DNA generally fluctuates more than histones, both in octasomes and in nucleosomes, and shows a tendency to unwrap from histone core at the DNA ends. In the HFD of 2x(CENP-A/H4)$_2$ octasomes, (CENP-A/H4)$_2$ dimers at the dyad (Fig S6A; chains C and E for CENP-A and D and F for H4) are more stable and comparable in fluctuations to dimers in 2x(H3/H4)$_2$ octasomes (Fig S6B), whereas (CENP-A/H4)$_2$ dimers near the DNA ends (Fig S6A; chains A and G for CENP-A and B and H for H4) are very dynamic. Interestingly, RMSF analysis shows that the larger motions occur at the H4/H4 four-helix bundles at the dyad of octasomes (which includes L2 loops) and not at the CENP-A/CENP-A of H3/H3 four-helix bundles at the SHL-3 and SHL3 positions (which also include L2 of these histones) (Fig 3A). This is consistent with rotation at the dyad (Fig S4A and B). In addition, great flexibility at the DNA level is also observed in octasomes compared with nucleosomes, with the largest amplitudes starting 50 bp away from the dyad. However, in nucleosomes (Fig S6C and D), the rigid histone core stabilizes the DNA regions that are in contact with the histones and show less variation than the regions exposed to the solvent. In contrast, both the histone core and the wrapped DNA in octasomes show high fluctuations that are more pronounced in 2x(CENP-A/H4)$_2$ octasomes compared with 2x(H3/H4)$_2$ octasomes.

To further analyze the concerted motions in octasomes (and nucleosomes), we performed principal component analysis (PCA). Indeed, the first eigenvector corresponds to an oscillatory movement of DNA ends around two hinge regions located at base pair positions (bps) −46 and +60 from the dyad in the 2x(CENP-A/H4)$_2$ octasome and at positions −39 and +60 from the dyad in the 2x(H3/H4)$_2$ octasome (yellow circles in Fig 3B). However, a similar movement was not observed in nucleosomes (data not shown). The DNA deformation is accompanied by a corresponding movement in the L1-L2 region between chains C/D of the octasomes, which is localized near the DNA hinge. This is also very consistent with the experimental data, which indicate multiple conformations after 52 bp from the dyad on one side and 62 bp on the other side, which could not be visualized in the cryo-EM maps even after careful particle classification.

The deformation of DNA is not limited to the flexible ends, but also to the separation between the two gyres of the DNA double helix. Therefore, we measured clamshell angle $\theta$ (theta), which is defined between the center of mass of the DNA base pair at the dyad and the base pairs in the middle between the dyad and the

DNA ends (Fig 3C). Here, we see that both types of nucleosomes have $\theta$ = 18° ± 1° during MD simulations, whereas 2x(H3/H4)$_2$ octasomes have $\theta$ = 26° ± 1° and 2x(CENP-A/H4)$_2$ octasomes have $\theta$ = 28° ± 1°. These values confirm that octasomes have a much larger gyre opening than nucleosomes and that the DNA gyres in 2x(CENP-A/H4)$_2$ octasomes are significantly further apart compared with 2x(H3/H4)$_2$ octasomes. A similar trend can be seen when measuring the distance between two gyres (Fig 3D; Table S3). It is noteworthy that both the distance between gyres and clamshell angle, although significantly different between the two octasome types, are still quite stable and result in a relatively sharp peaks in the probability plot.

Overall, the MD analysis confirms that 2x(CENP-A/H4)$_2$ octasomes are highly dynamic but stable molecular structures.

## (CENP-A/H4)$_2$ tetrasomes and octasomes bind CENP-C with reduced affinity

We wanted to understand whether (CENP-A/H4)$_2$ tetrasomes and 2x(CENP-A/H4)$_2$ octasomes, as possible chromatin units, are able to recognize and bind CENP-A binding proteins, CENP-C and CENP-N.

It is known from previous structural work that two regions of human CENP-C (central binding domain, CENP-C$^{CR}$ and CENP-C motif, CENP-C$^{motif}$) bind CENP-A nucleosomes in a similar manner, with CENP-C$^{CR}$ having higher affinity and specificity in the context of human proteins (8, 18, 33). When we incubate (CENP-A/H4)$_2$ tetrasomes with CENP-C$^{CR}$, we see clear binding (Fig 4A; the band corresponding to CENP-A tetrasomes disappears). In contrast, when (H3/H4)$_2$ tetrasomes are incubated with CENP-C$^{CR}$, CENP-C$^{CR}$ binds to the free DNA to a higher extent than to H3 tetrasomes (Fig 4A). The band corresponding to free DNA disappears, but the band corresponding to tetrasomes shows slightly decreased intensity with the addition of CENP-C$^{CR}$. When 2x(CENP-A/H4)$_2$ octasomes are incubated with CENP-C$^{CR}$, we see binding for octasomes (the octasome band disappears), but unlike CENP-A nucleosome/CENP-C$^{CR}$ complexes, which migrate as defined bands on the gel (Fig S7A), the binding of CENP-C$^{CR}$ to octasomes results in a smear on the gel.

The binding of CENP-C$^{CR}$ by (CENP-A/H4)$_2$ tetrasomes and octasomes with lower affinity compared with CENP-A nucleosomes is consistent with previous high-resolution structures of CENP-C$^{CR}$/CENP-A nucleosome complexes (8, 11, 12, 13, 16, 19, 20). There, the binding is controlled by two interaction surfaces. One involves hydrophobic interactions between aromatic and hydrophobic residues in CENP-C$^{CR}$ and the C-terminal part of CENP-A, and the other interaction interface involves electrostatic interactions between two arginine residues in CENP-C and the acidic residues on H2A (Fig S7A). Although both 2x(CENP-A/H4)$_2$ octasomes and (CENP-A/H4)$_2$ tetrasomes have a hydrophobic C-terminal part of CENP-A available for interaction with CENP-C$^{CR}$, both lack the electrostatic part of the interactions provided by H2A. This would explain why CENP-A–containing complexes can still bind CENP-C$^{CR}$, albeit with lower affinity compared with CENP-A nucleosomes.

octasomes and 36 and 114 on the CENP-A nucleosome. **(E)** Same as in (D). Highlighted in black are the $\alpha$N helices of CENP-A (left) and H3 (middle), which are disordered and thus invisible in our map but are "pushing away" the two halves of the octasomes allowing for a more dynamic structure. On the right, the beta-sheet formed between H4 and H2A holds the CENP-A nucleosome (right) together to form a more compact particle.

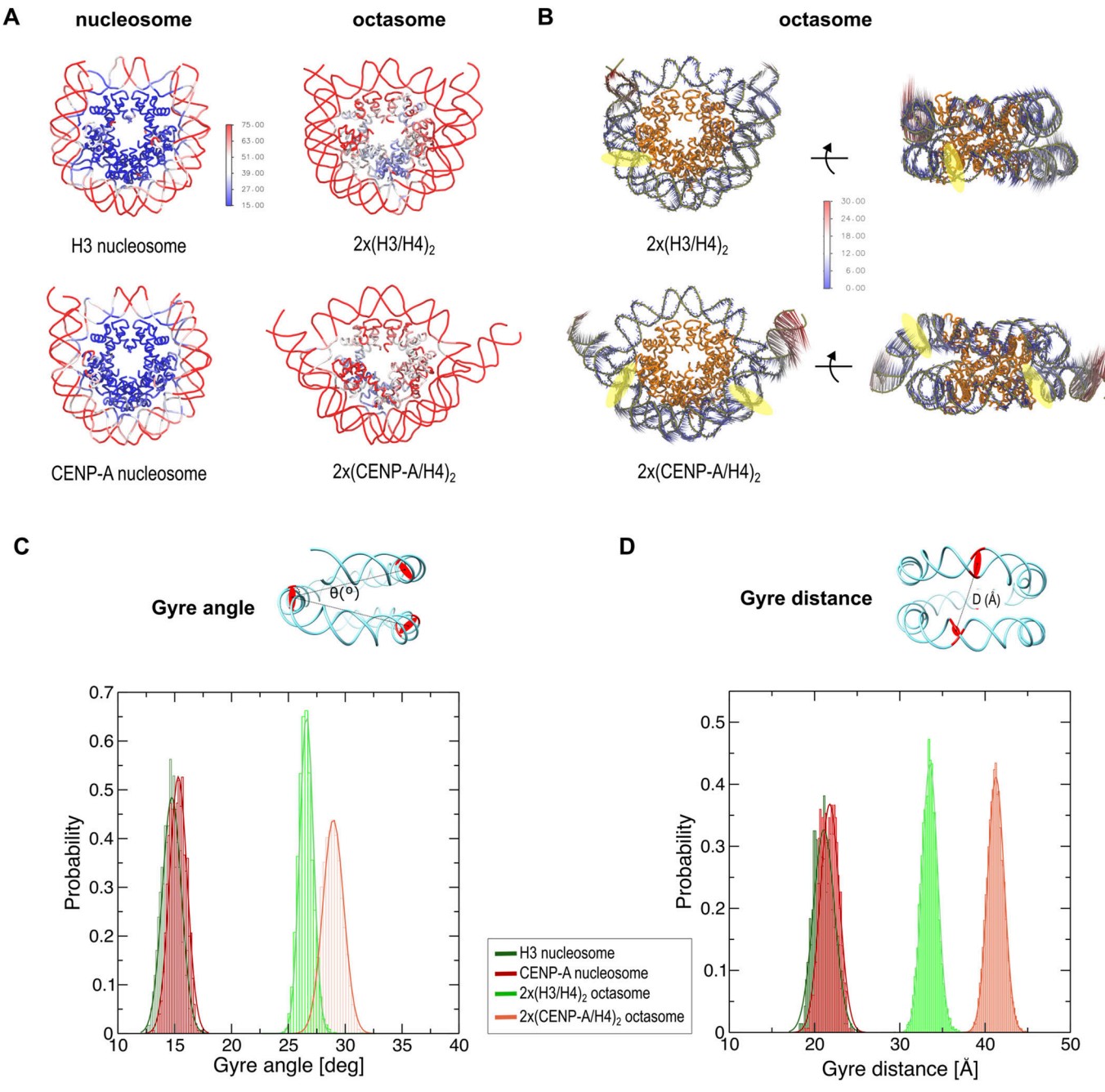

**Figure 3. Molecular dynamics of 2x(CENP-A/H4)$_2$ octasomes.**
**(A)** Root mean square fluctuations (RMSF) observed during MD simulations for nucleosome and octasomes are shown as a heat map on the ribbon structures of nucleosomes and octasomes. The residues are colored according to their thermal B-factors (blue—low fluctuations; red—high fluctuations) predicted by MD. **(B)** Porcupine diagram illustrates the vectors of atomic displacements on the ribbon diagram of octasomes. The histone core is yellow; the DNA is gray. The arrows represent the directions and magnitudes of movement. The color code correlates with the size of the displacement in Å. **(C, D)** Probability plots showing the distribution range of the opening angles (C) and the opening distances (D) of the DNA gyres for the octasome and nucleosome structure observed in the MD simulation. The DNA base pairs (0, +39, −39) used to calculate the angles/distances are highlighted by red circles in the cyan ribbon diagram of DNA in 2x(CENP-A/H4)$_2$ octasomes at the top of the plots.

## (CENP-A/H4)$_2$ tetrasomes and octasomes bind N-terminal part of CENP-N

Although the binding of CENP-N to CENP-A was first demonstrated 15 yr ago (21), our understanding of this interaction has evolved with numerous high-resolution structures. Initial biophysical experiments and crystal structures involving only the CENP-A nucleosome and N terminus of CENP-N, CENP-N$^{N-term}$, indicated that CENP-N specifically recognizes the $^{80}$ArgGly$^{81}$ insertion (CENP-A$^{RG-loop}$) in histone CENP-A, which is absent in its

## A  (CENP-A/H4)₂ tetrasome and octasome bind CENP-C^CR

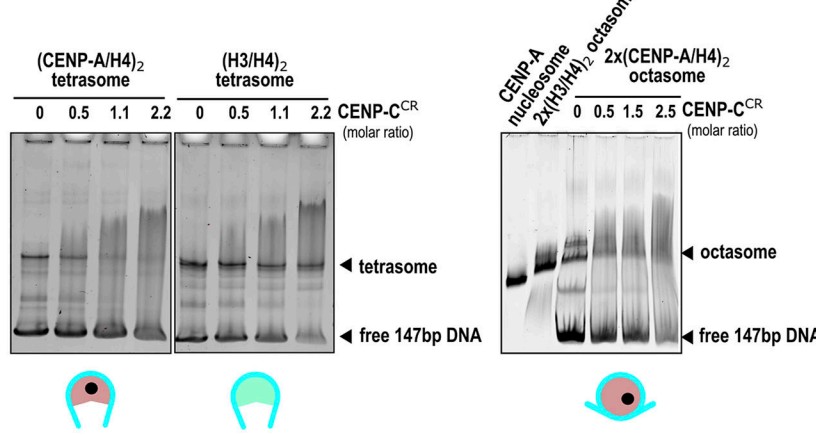

## B  CENP-N/L engages CENP-A substrates but not H3

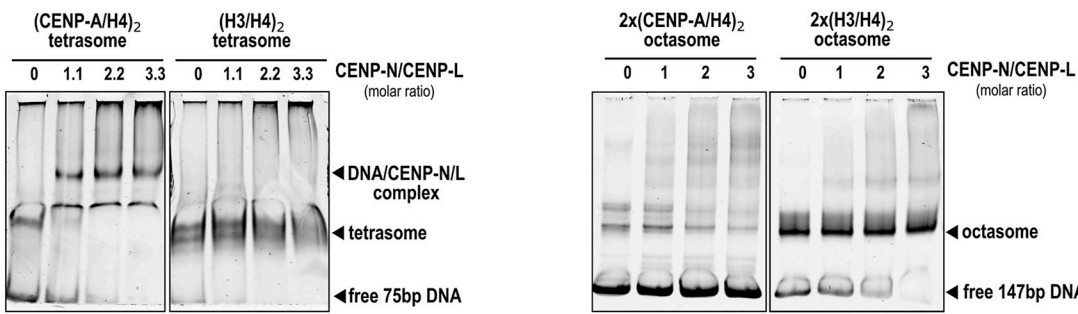

**Figure 4.  Native-gel assessment of CCAN component interaction with CENP-A assemblies.**
**(A)** CENP-C binding. 5% native PAGE shows that both the (CENP-A/H4)₂ tetrasome (left) and the 2x(CENP-A/H4)₂ octasome (right) interact with CENP-C^CR. Complexes migrate as broadened/smeared signals under these conditions, in contrast to the CENP-A nucleosome + CENP-C complex, which yields a discrete band (see Fig S7A). Schematic icons beneath the gels indicate particle types; CENP-A cores are red, H3 cores are green, DNA is cyan, and the CENP-A C-terminal tail is marked by a black dot. Note that the H2A acidic patch is absent in both tetrasome and octasome. **(B)** CENP-N/L binding. 5% native PAGE comparing CENP-N/L with CENP-A versus H3 assemblies. CENP-N/L engages CENP-A–containing substrates, whereas H3 controls show no detectable binding under the same conditions. Qualitatively, tetrasome lanes display earlier/more pronounced CENP-N/L–dependent mobility changes than octasome lanes, indicating different apparent engagement/kinetics in this assay. Source data are available for this figure.

canonical counterpart, H3. The CENP-A^RG-loop is exposed on the surface of CENP-A nucleosomes and is recognized and bound (together with the adjacent DNA) by several CENP-N residues (reviewed in reference 18). However, full-length CENP-N forms a complex with CENP-L (34), and structures of CENP-A nucleosomes obtained in the presence of the full set of CCANs with fission yeast and human proteins, including the CENP-N/L complex, show that the CENP-N/L complex binds DNA extending from the nucleosome without specifically interacting with any histones in the nucleosomes. Moreover, the conformation that the CENP-N/L complex adopted in these structures is incompatible with the interaction involving the CENP-A^RG-loop in the context of the CENP-A nucleosome as observed in the CENP-A nucleosome/CENP-N^N-term structures (18). Different binding modes of CENP-N have raised the question of alternative arrangements between CENP-A and CCANs that could characterize different phases of the cell cycle (14). With this in mind, we tested whether (CENP-A/H4)₂

tetrasomes and octasomes can bind CENP-N^N-term and the CENP-N/L complex.

First, we examined the binding of the N-terminal part of human CENP-N, CENP-N^N-term to (CENP-A/H4)₂ tetrasomes and octasomes. We found that the band for 2x(CENP-A/H4)₂ octasomes disappears from the gel even when substoichiometric amounts of CENP-N^N-term are added and several other bands are formed, which are likely to be complexes of CENP-N^N-term with DNA, tetrasomes, and octasomes (Fig S7B, left gel). Interestingly, the binding of CENP-N^N-term to the CENP-A nucleosome always leads to the formation of multiple laddered bands, indicating the formation of a CENP-A nucleosome/CENP-N^N-term complex and its further oligomerization by CENP-N^N-term (Fig S7B, right gel). This phenomenon was visualized before (24), but it is currently unclear whether the CENP-N–induced oligomerization has a physiological role in cells. Interestingly, we see that 2x(H3/H4)₂ octasomes could also bind CENP-N^N-term and generate a band ladder on the native gel (Fig S7B, middle gel).

## (CENP-A/H4)₂–containing assemblies bind the CENP-N/L complex

Because CENP-L, a binding partner of CENP-N, is also always present in centromeres, we tested whether (CENP-A/H4)$_2$ tetrasomes and octasomes can bind the CENP-N/L complex (Fig 4B). In native-gel assays, CENP-N/L engaged CENP-A–containing assemblies (tetrasome and octasome) but not H3 controls under matched conditions. In side-by-side titrations, tetrasome lanes exhibited CENP-N/L–dependent mobility changes (smearing) at lower protein:DNA ratios than octasome, indicating different apparent engagement/kinetics in this qualitative assay. Complexes with tetrasome and octasome appeared heterogeneous/smeared, whereas the sharp band observed at high CENP-N/L corresponds to DNA·CENP-N/L. We also attempted to visualize the (CENP-A/H4)$_2$/CENP-N/L complex by cryo-EM but could only obtain low-resolution density maps so far.

Finally, we asked whether (CENP-A/H4)$_2$ tetrasomes could bind CENP-HIKM directly or upon binding CENP-N/L, but the results were not conclusive.

# Discussion

Since the discovery that the position of the centromere on the chromosomes is dominated by an epigenetic rather than a genetic component (35), great efforts have been made to understand the molecular determinants of this phenomenon. In the following years, numerous experiments identified the histone H3 variant CENP-A as a key protein component that is necessary and sufficient for the establishment of functional centromeres (reviewed in reference 6). However, only recently, thanks in part to rapid technological advances in cryo-EM, the high-resolution structure of the CENP-A nucleosome in complex with CCANs from human and yeast has been published (11, 12, 13).

Surprisingly, however, there are no extensive contacts between the protein part of the CENP-A nucleosome and the rest of the CCAN complex in the CENP-A/CCAN structure of *S. pombe* and humans. The only CCAN component that interacts with CENP-A is CENP-C, which remains unstructured, and thus "invisible," in the cryo-EM structure in the sections immediately before and after interaction with CENP-A (Figs 5A and S9). Furthermore, the same nucleosome/CCAN structure can also assemble without CENP-A, that is, on naked DNA (10), or with the H3 nucleosome (13), so the question of how exactly CENP-A specifically recruits CCAN is therefore still open.

Here, we present the structure of 2×(CENP-A/H4)$_2$ octasome that can form in chromatin when H2A/H2B is absent. The architecture is highly similar to the recently reported CENP-A/H4 octasome (29) and parallels earlier 2×(H3/H4)$_2$ "H3–H4 octasomes" observed by cryo-EM and in-cell cross-linking (28). Additional observations support the plausibility of H3/H4-only particles in vivo: expression of only H3 and H4 can package the *E. coli* genome (36), and cryo-electron tomography of interphase nuclei reveals more open, variably wrapped chromatin particles (37). In our preparations, densely packed 2×(CENP-A/H4)$_2$ octasomes also form looser, flexible fibers

reminiscent of archaeal-like histone polymers (31, 32, 38), suggesting a packing mode that could contribute to centromere-specific chromatin features.

Functionally, our native-gel assays indicate that CENP-N/L engages CENP-A–containing assemblies, whereas H3 controls show no detectable binding. Engagement is more readily detected for tetrasomes than for octasomes under matched conditions, consistent with greater RG-loop accessibility when a second DNA wrap is absent (Fig S8A and B). In contrast, CENP-C binds the CENP-A nucleosome robustly but interacts only weakly with CENP-A/H4 assemblies lacking the H2A acidic patch. Together with the octasome geometry (this work and reference 29), these data suggest that accommodating CCAN on octasome RG-loop sites would require conformational rearrangement and/or local DNA unwrapping, whereas (CENP-A/H4)$_2$ tetrasomes provide a CCAN-receptive intermediate via the RG loop.

We therefore propose a cell cycle–coupled alternation in which tetrasome/octasome states prevail immediately after CENP-A/H4 deposition in G1 and mature to the CENP-A nucleosome as H2A/H2B is incorporated in the S phase, with two illustrative CCAN-binding modes: RG-loop capture via CENP-N on subnucleosomal CENP-A/H4 assemblies and linker-DNA anchoring via CENP-C on the nucleosome (Figs 5 and S9), consistent with prior proposals of kinetochore plasticity (19, 39, 40, 41). This perspective raises the question of how newly deposited CENP-A replaces placeholder histones and how stable subnucleosomal CENP-A/H4 cores are maintained, which we address next by considering the H3.3 placeholder model and the timing of CENP-A loading.

It has been proposed that H3.3 is deposited at centromeres during the S phase as a "placeholder," later replaced by CENP-A at the mitotic exit (42). How H3.3 is targeted within centromeric chromatin, and whether the G1 replacement entails exchange of a whole nucleosome or subnucleosomal complexes, remains unsettled. Consistent with this latter possibility, reference 43 reported that CENP-A loads with H4 in early G1 in the absence of H2A/H2B, and that the resulting CENP-A/H4 subnucleosomal core shows unusual persistence at centromeres across multiple cell cycles. The CENP-A targeting domain (CATD)—comprising the L1 loop (including the RG loop) and α2 helix—is required for HJURP binding and centromere targeting (44, 45), yet the long-term chromatin stability observed by reference 43 does not depend on HJURP. This stability is plausibly supported by the strong hydrophobic CENP-A–H4 interface (25, 44) and, in our model, by early CCAN engagement in which newly incorporated (CENP-A/H4)$_2$ tetrasomes interact with CENP-N/L via the CENP-A RG loop, thereby reinforcing retention without invoking broader claims about other histones.

Indeed, we constructed an AlphaFold model that has the (CENP-A/H4)$_2$ tetrasome as a base and harbors CCAN through the CENP-A$^{RG-loop}$ interaction (Figs 5B and S9), and we see no steric clashes. Moreover, CCAN wraps tightly around the (CENP-A/H4)$_2$ tetrasome in our model, shielding it from other possible interactions. In this way, the stability of CENP-A/H4 and CCAN at the centromere would be ensured after mitosis and before the S phase, when new CENP-N is loaded (46) and the CENP-A nucleosome is completed by the incorporation of H2A/H2B. These structural changes in the S phase would trigger the translocation of CCAN to nucleosomal DNA

## A    CENP-A nucleosome with CCAN anchored throug CENP-C (PDB 7YWX)

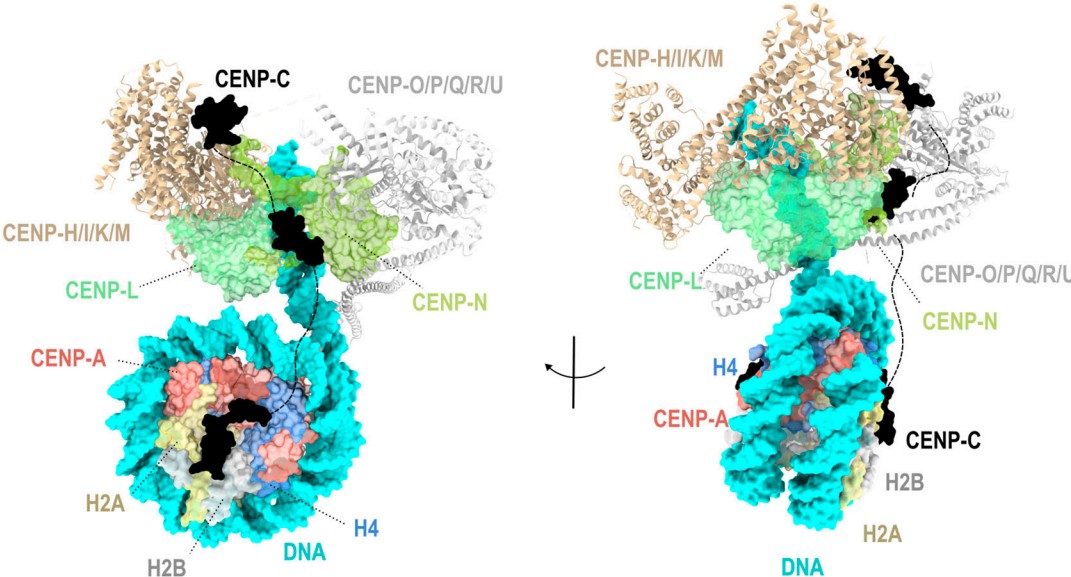

## B    CENP-A tetrasome with CCAN anchored through CENP-N (this work+AlphaFold)

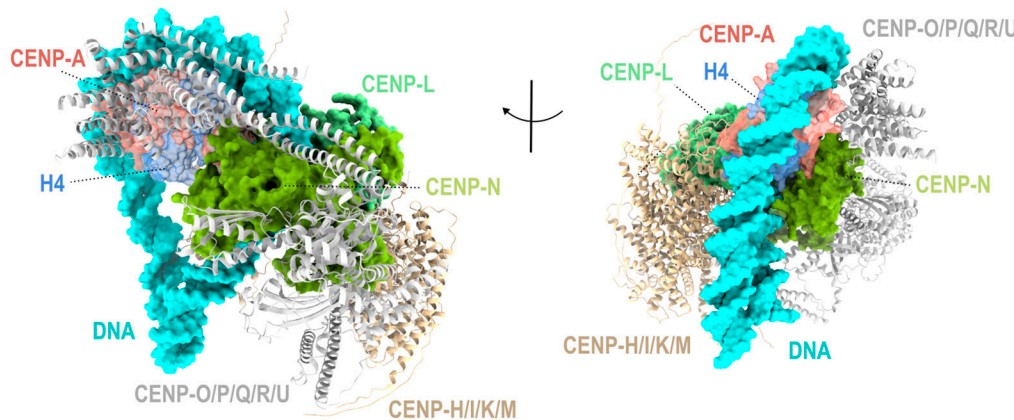

**Figure 5.    Different models for CCAN engagement with CENP-A chromatin.**
**(A)** Space-filling representation of CCAN bound to the CENP-A nucleosome generated using PDB 7YWX. **(B)** Space-filling representation of CCAN bound to CENP-A tetrasome generated using our structure of CENP-A tetrasome and AlphaFold modeling for the CCAN complex. The structure was minimized and shows no steric clashes. CENP-A is red, H4 is blue, H2A is yellow, H2B is gray, DNA is cyan, CENP-C is black, CENP-N-L is green, CENP-H-I-K-M is beige, and CENP-O-P-Q-U-R is light gray. CENP-H-I-K-M and CENP-O-P-Q-U-R are shown as ribbons for clarity.

(possibly assisted by remodeler (47)), where it remains anchored by CENP-C and ready to recruit the kinetochore in mitosis. The model also provides an explanation for the central role of CENP-N in maintaining CCAN integrity during the cell cycle (46, 48, 49).

Although theoretically possible, we were unable to successfully assemble the (CENP-A/H4)$_2$–tetrasome–CCAN complex in vitro with recombinantly purified CCAN, suggesting that such a structure, if indeed present in the cell, would need to be further stabilized by

components missing from our in vitro assembly or by possible posttranslational modifications of the CCAN components and/or CENP-A. We also found that 2x(CENP-A/H4)$_2$ octasomes readily convert to more stable nucleosomes in the presence of free H2A/H2B (Fig S10), which would complicate the isolation of such particles from cells and would be consistent with a previous detailed study in which full octameric homotypic CENP-A nucleosomes were found present throughout the cell cycle upon chromatin isolation (27).

Although we are aware that our model is speculative in the absence of direct cell-based evidence, we believe that our results will motivate further research to answer two important questions: 1. How are the centromere proteins organized immediately after CENP-A loading in the G1 phase? And 2. when and how does the specific interaction between the CENP-A$^{RG-loop}$ and CENP-N/L take place during the cell cycle? We believe that the further development of imaging techniques such as cryo-electron tomography, which allow visualization of centromeres in natural environment (50 Preprint) and the isolation of native kinetochores (51), is crucial to obtain an accurate and detailed picture of centromere organization during the cell cycle.

# Materials and Methods

### Protein cloning and purification

**(H3/H4)$_2$ and (CENP-A/H4)$_2$** were expressed as heterotetramers. (H3/His-H4)$_2$ and (His-CENP-A/H4)$_2$ were each cloned into pST44 plasmids and bicistronically expressed in *E. coli* Rosetta 2 cells overnight at 18°C. Harvested cells were resuspended in lysis buffer (50 mM Tris, pH 7.5, 2 M NaCl, 30 mM imidazole, 5% glycerol, 5 mM BME [β-mercaptoethanol]) and sonicated. The supernatant was injected onto a 5-ml HisTrap HP column (Cytiva), and (H3/His-H4)$_2$ or (His-CENP-A/H4)$_2$ was eluted with buffer containing 20 mM Tris, pH 7.5, 2 M NaCl, 5% glycerol, 5 mM BME, and 300 mM imidazole. The His tag was cleaved overnight with in-house–prepared TEV protease, and the sample was subjected to a final cation-exchange chromatography step (HiTrap SP HP 5 ml; Cytiva). Elution was performed in steps with the final buffer containing 20 mM Tris, pH 7.5, 2 M NaCl, and 2 mM DTT. The purity of the histones was assessed using SDS–PAGE gel (Fig S1D).

### CENP-N$^{1–240}$

CENP-N$^{1–240}$ with a His tag on the C terminus was expressed overnight at 18°C in *E. coli* pLysS. Pelleted cells were dissolved in 50 mM NaPO$_4$, pH 8, 300 mM NaCl, 10 mM imidazole, 0.1% Tween-20, and lysed by sonication. CENP-N$^{1–240}$ was purified using a 5-ml HisTrap FF column, and eluted with 50 mM NaPO$_4$, pH 7.5, 500 mM NaCl, 300 mM imidazole, 20% glycerol. The eluted protein was further purified on a Superdex S200 16/600 size-exclusion column preequilibrated with 50 mM NaPO$_4$, pH 7.5, 500 mM NaCl, 20% glycerol.

### CENP-N/L

The human CENP-N/His-CENP-L complex was expressed in Sf9 cells for 3 d at 27.5°C (Lund Protein Production Platform) as described before (48). Harvested cells were resuspended in buffer containing 20 mM Tris, pH 7.5, 0.3 M NaCl, 5% glycerol, 5 mM BME, 0.05% Tween-20, EDTA-free anti-protease cocktails (Sigma-Aldrich), and the sample was sonicated. The supernatant was injected onto a 5-ml HisTrap HP column (Cytiva), and CENP-N/L complex was coeluted using 20 mM Tris, pH 7.5, 0.3 M NaCl, 5% glycerol, 5 mM BME, and 300 mM imidazole. A further size-exclusion chromatography

purification step was performed using a S200 column (Cytiva) preequilibrated in 20 mM Hepes 7.5, 300 mM NaCl, 2 mM DTT. The purity of the complex was assessed using SDS–PAGE gel (Fig S1D).

### CENP-C

CENP-C$^{426-537}$ was expressed in *E. coli* pLysS and purified as previously described in reference 52. Briefly, GST-tagged CENP-C$^{426-537}$ was purified on a glutathione column (GSTrap HP Cytiva). GST was subsequently cleaved overnight by PreScission protease and separated from CENP-C using cation-exchange chromatography (HiTrap SP HP 5 ml; Cytiva). The purity of the protein was assessed using SDS–PAGE gel (Fig S1D).

### DNA preparation

145-bp and 200-bp 601 superpositioning DNAs and 147-bp α-satellite DNA were purified as described in reference 52, 53. Briefly, HB101 cells transformed with pUC57 plasmids containing 6 × 147-bp α-satellite DNA or 8 × 145-bp 601 or 19 × 200-bp 601 superpositioning DNA (gift from Ben Black, UPenn) were grown at 37°C overnight, and DNA was extracted with phenol/chloroform. Plasmids were digested using restriction enzymes and further purified using anion-exchange chromatography (resource Q 6 ml column; Cytiva).

601 (145 bp): ATCAGAATCCCGGTGCCGAGGCCGCTCAATTGGTCGTAG ACAGCTCTAGCACCGCTTAAACGCACGTACGCGCTGTCCCCCGCGTTTTAA CCGCCAAGGGGATTACTCCCTAGTCTCCAGGCACGTGTCAGATATATA CATCGAT.

601 (200 bp): TATGTGATGGACCCTATACGCGGCCGCCCTGGAGAATCC CGGTGCCGAGGCCGCTCAATTGGTCGTAGCAAGCTCTAGCACCGCTTAAAC GCACGTACGCGCTGTCCCCCGCGTTTTAACCGCCAAGGGGATTACTCCCTA GTCTCCAGGCACGTGTCAGATATATACATCCTGTGCATGTATTGAACAGCG ACTCGGGT.

α-satellite (75 bp): TTTGGAAACTGCTCCATCAAAAGGCATGTTCAGCTC TGTGAGTGAAACTCCATCATCACAAAGAATATTCTGAGA.

α-satellite (147 bp): ATCAAATATCCACCTGCAGATTCTACCAAAAGT GTATTTGGAAACTGCTCCATCAAAAGGCATGTTCAGCTCTGTGAGTGAAAC TCCATCATCACAAAGAATATTCTGAGAATGCTTCCGTTTGCCTTTTATATG AACTTCCTCGAT.

di-α-satellite (342 bp): CTGAGGCCTGTGGTAGTAAAGGAAAGAACTTCA TATAAAAACTAGACGGTAGCACCCTCAGAAAATTCTTTGTGACGATGGAGTTTA ACTCAGAGAGCTGAACATTCGTTATGATGGAGCAGTTTCCAAACACACGTT TTGTAGAATCTGCAAGGGGATATTTGGACCTTCCGGAGGATTTCGTTGGA AACGGGATCAACTTCCCATAACTGAACGGAAGCAAACTCAGAACATTCTTT GTGATGTTTGTATTCAACTCACAGAGTTGAACCTTCCTTTGATAGTTCAGG TTTGCAACACCCTTGTAGTAGAATCTGCAAGTGTATATTTTGACCACTT TGG.

### Octasome and poly-tetrasome complex assembly

Octasome and poly-tetrasome complexes were assembled using salt gradient dialysis. (CENP-A/H4)$_2$ or (H3/H4)$_2$ hetero-tetramers were mixed with 601 (145-bp) or α-satellite (147-bp) DNA sequences in a molar ratio of 2:1 for octasome, and with di-α-satellite (342-bp) DNA sequence in a molar ratio of 5:1 for poly-tetrasome in high salt buffer (20 mM Tris, pH 7.5, 2 M NaCl, 2 mM DTT). Gradient

dialysis to low salt buffer (10 mM Tris, pH 7.5, 100 mM NaCl, 2 mM DTT) was performed overnight at a flow rate of 1.5 ml/min using a dual-channel peristaltic pump. The next day, complexes were dialyzed against 10 mM MOPS 7.5, 100 mM NaCl, 2 mM DTT buffer and stored at 4°C, if necessary. The complex quality was checked using a 5% native PAGE gel.

## Binding assays

1 $\mu$M of 2x(CENP-A/H4)$_2$ or 2x(H3/H4)$_2$ octasome assembled on 601 (145- or 200-bp) DNA or $\alpha$-satellite DNA was mixed with various amounts of CENP-C$^{426–537}$, CENP-N$^{1–240}$, CENP-N/L, or CENP-H/I$^{57-C}$/K/M and incubated for 1 h on ice. Complex formation was monitored using a 5% native PAGE gel.

## MNase experiments

0.5 $\mu$g of nucleosome, octasome, or tetrasome complexes (based on DNA concentration), assembled on 147-bp $\alpha$-satellite DNA, was incubated with 1 Kunitz unit of micrococcal nuclease (NEB) in a buffer containing 10 mM Tris–HCl, pH 7.5, 3 mM CaCl$_2$, and 1 mM DTT at room temperature. Reactions were quenched at different time points (2, 5, 8, 10, and 15 min) by adding 250 $\mu$l PB buffer (QIAGEN QIAquick PCR Purification Kit) supplemented with 10 mM EGTA. DNA from each sample was purified using the QIAquick PCR Purification Kit, and the extent of DNA digestion was quantified using the 2100 Bioanalyzer (Agilent). All experiments were performed in duplicate.

## Cryo-EM grid preparation, and data acquisition and processing

The octasome complexes were cross-linked with 0.05% glutaraldehyde for 40 min on ice. 3 $\mu$l of the cross-linked 2x(H3/H4)$_2$ octasome complex assembled on 147-bp $\alpha$-satellite DNA (cross-linking was not quenched, but rather the sample was immediately applied on the grid and plunged in liquid ethane) and uncross-linked 5x (CA/H4)$_2$ poly-tetrasome complex assembled on 342-bp $\alpha$-satellite DNA at a concentration of 0.8–1 mg/ml was applied to freshly glow-discharged Quantifoil R2/1 300 mesh grids. Cross-linked 2x(CA/H4)$_2$ octasome assembled on 147-bp $\alpha$-satellite DNA was applied on graphene oxide–coated grids at a concentration of 0.1 mg/ml. All grids were blotted for 5 s and frozen in liquid ethane using a FEI Vitrobot automatic plunge freezer. Humidity in the chamber was maintained at 100%.

2x(H3/H4)$_2$ and 2x(CENP-A/H4)$_2$ octasome datasets were acquired using the Titan Krios electron microscope (Thermo Fisher Scientific) at 300 kV (cryo-EM facility at UCEM, Umeå University, Sweden). Both datasets, 14,209 movies for 2x(H3/H4)$_2$ octasome particles and 12,627 for 2x(CENP-A/H4)$_2$, were acquired using a Gatan Summit K2 electron detector at a magnification of 165 k and a pixel size of 0.82 Å. The total electron exposure of 57.5 e-/Å2 was distributed over 40 frames. (CENP-A/H4)$_2$ poly-tetrasome dataset was acquired at UCEM cryo-EM Facility using the Titan Krios electron microscope (Thermo Fisher Scientific) at 300 kV. 8,140 movies were acquired using the Falcon 4i electron detector at a magnification of 165 k and a pixel size of 0.704 Å. The total electron exposure of 50 e-/Å2 was distributed over 828 frames (Fig S5A).

Data were acquired using EPU (Thermo Fisher Scientific) automated data acquisition software with AFIS. The defocus range was from –1 to –3.8 $\mu$m with a step size of 0.3 $\mu$m.

Movie frames were aligned using patch motion in CryoSPARC 3.1 (54). CTF was estimated in a patch manner, and several hundreds of particles were manually picked. The resulting useful particles were then used for automatic particle picking using Topaz (55). The 2D class averages were generated in CryoSPARC 3.1 (Figs S2A, S3C, and S5B). Inconsistent class averages were removed from further data analysis. The 3D classifications and refinements were subsequently done in CryoSPARC 3.1 (Figs S2B and S3D). The initial reference was generated using ab initio and was filtered to 30 Å, and C1 symmetry was applied during homogeneous refinements. Particles were split into two datasets and refined independently, and the resolution was determined using the 0.143 cutoff (Fig S2E). The Euler angle distribution of particles used in the 3D reconstruction was visualized in Chimera (Fig S2D). Local resolution was determined with local resolution estimation (Figs S2C and S3E and F). All maps were filtered to local resolution using CryoSPARC 3.1 with a B-factor determined in the refinement step.

Statistics on cryo-EM maps are summarized in Tables S1 and S2.

## Model building

The model was built in Coot (56) and refined using Phenix real_space_refine (57). Figs are prepared with Chimera (58).

## Molecular dynamics

### System setup

Molecular mechanics models for the 2x(H3/H4)$_2$ or 2x(CENP-A/H4)$_2$ octasomes were built starting from the experimental cryo-EM structures introduced in this work. As the experimental resolution allows for characterization of only the core protein regions of the complexes, models of the complete histones were created by reconstructing the missing residues with the MODELLER package (59). Missing DNA ends not wrapped to the histones were added as canonical double-helix moieties, using the web 3DNA server (60). For comparison, we also simulated conventional centromeric (PDB 6O1D) (61) and noncentromeric nucleosomes (PDB 3LZ0) (62) on 601 superpositioning sequence (63), using 3D models from, respectively, cryo-EM data (3.4 Å resolution) and x-ray diffraction (2.5 Å resolution) as starting coordinates. All missing hydrogens were added using geometric restraints. The protonation states of all titratable residues were determined at pH 7, considering possible pKa shifts (64, 65). All the systems were solvated by TIP3P waters (66) by setting the initial edge of the cubic periodic box to a length of 20 nm. Sodium and chloride ions were added to the system to achieve charge neutralization, and a saline concentration of 150 mM, mimicking physiological ionic strength. The resulting systems contained roughly 0.8 million of atoms (Table S3). Interaction energy terms were described using the AMBER force field (67, 68), electrostatic interactions were computed using the particle mesh Ewald method (69), and Lennard–Jones (LJ) potentials mimicking van der Waals interaction were approximated by cutoff scheme. A

cutoff radius of 1.2 nm was used for both the LJ and the short-range electrostatics terms, updating the pair list every 20 molecular dynamics (MD) steps using a Verlet cutoff scheme (70). All simulations were run for 900 ns.

### Simulation parameters

All the systems underwent several rounds of steepest descent energy minimization (maximum 10,000 cycles) and thermal equilibration by simulated annealing in the NPT to reach the target temperature of 300 K, targeted by the canonical velocity rescaling algorithm (71), using a coupling constant $\tau_T$ = 1.0 ps. Constant pressure of 1.0 Bar was achieved by coupling MD simulations to a stochastic cell rescaling barostat (72), using a coupling constant $\tau_p$ = 2.0 ps. The timestep for all the simulations was set to $\Delta t$ = 2 fs, using the leap-frog integrator. The LINCS algorithm (73) was used to constrain all the bonds involving hydrogen atoms to their equilibrium distance. All simulations were performed using the GROMACS software (74).

### Minimization of the AlphaFold models

After obtaining tetrasome/CCAN complex models using AlphaFold, the protonation states of all titratable amino acids at neutral pH were determined by calculating their pKa values with PDB 2PQR (75). The structures were then energy-minimized using GROMACS software (76) and the AMBER99-bsc1 force field parameters (67, 68). Van der Waals interactions were calculated with a 1.0-nm cutoff distance, whereas electrostatic interactions were calculated using the particle mesh Ewald (PME) method (77). The steepest descent energy minimization was performed for up to 5,000 steps, aiming for a convergence threshold with a force constant of 1,000 kJ/(mol nm). Hydrogen bonds were constrained by the LINCS algorithm.

### Measuring angle and distance for gyre opening

The gyre opening distance and angles were defined considering the base pair of the wrapped DNA at the dyad location (base-pair residues numbered "0" in the 6O1D and octasome PDBs; base-pair residues numbered "73" in the 3LZ0 structure), and the base pairs at the opposite location on the two DNA arms (±39 in the DNA sequence).

The gyre distance and angles were defined as the distance between the N1 atoms of the purine nucleobase at ±39 positions from the dyad, whereas the gyre angle was defined as the angle formed by the N1 atoms of the purine nucleobases at dyad-39, dyad, dyad+39 positions.

Measured angles and distances are summarized in Table S4.

## Data Availability

The atomic coordinate model for 2x(CENP-A/H4)$_2$ octasome has been deposited in the PDB with accession code 9GXA. Cryo-EM maps have been deposited in the EMBD with the accession codes: EMD-51645, EMD-51646, EMD-51647, EMD-51656. Molecular dynamics data for octasomes are available at reference 78 and for nucleosomes at reference 79.

## Supplementary Information

## Acknowledgements

We thank Prof. Lars Jansen (University of Oxford) for discussions at the beginning of the project phase and all members of the Sekulić group for their support and discussions throughout the project. N Sekulić and A Ali-Ahmad are supported by the NCMM and the Research Council of Norway (grant numbers 187615 and 325528). M Mors, X Li, M Carrer, and M Cascella were supported by the Research Council of Norway through the Centre of Excellence Hylleraas Centre for Quantum Molecular Sciences (grant number 262695), and the Norwegian Supercomputing Program (NOTUR) for computing hours and storage facilities (grants numbers NN4654K and NS4654K). Work in the Halic laboratory is funded by St. Jude Children's Research Hospital, the American Lebanese Syrian Associated Charities, and NIH awards 1R01GM135599 and 1R01GM141694. The cryo-EM data were collected at the Umeå Core Facilities for Electron Microscopy, a node of the Cryo-EM Swedish National Facility funded by the foundations of Knut and Alice Wallenberg, Erling Persson, and the Kempe family, SciLifeLab, Stockholm University, and Umeå University.

### Author Contributions

A Ali-Ahmad: conceptualization, data curation, formal analysis, validation, investigation, visualization, methodology, and writing—original draft, review, and editing.
M Mors: formal analysis, visualization, and methodology.
M Carrer: formal analysis and visualization.
X Li: data curation, formal analysis, investigation, visualization, and methodology.
S Bilokapić: conceptualization, investigation, and methodology.
M Halić: conceptualization and project administration.
M Cascella: formal analysis, supervision, and project administration.
N Sekulić: conceptualization, resources, supervision, funding acquisition, visualization, project administration, and writing—original draft, review, and editing.

### Conflict of Interest Statement

The authors declare that they have no conflict of interest.

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
