## [Reviewer comments · Life Science Alliance]

Structure and Dynamics of CENP-A/H4 Octasome Reveal a possible Intermediate in Centromeric Chromatin

Ahmad Ali-Ahmad, Mira Mors, Manuel Carrer, Xinmeng Li, Silvija Bilokapic, Mario Halic, Michele Cascella, and Nikolina Sekulić
DOI: <https://doi.org/10.26508/lsa.202503377>

Corresponding author(s): Nikolina Sekulić, University of Oslo and Ahmad Ali-Ahmad, University of Oslo

Review Timeline:

Submission Date:	2025-05-02
Editorial Decision:	2025-05-07
Revision Received:	2025-10-27
Editorial Decision:	2025-11-20
Revision Received:	2025-11-26
Accepted:	2025-12-03

Scientific Editor: Tim Fessenden

Transaction Report:

Please note that the manuscript was previously reviewed at another journal and the reports were taken into account in the decision-making process at *Life Science Alliance*.

Referee #1 Review

Report for Author:

This is a useful paper that deserves publication (at least in some reasonable journal), but not in its present form.

When the first high-resolution structure of a nucleosome appeared, it was immediately evident that the histone dimers formed a quasi-continuous helical ramp. "Quasi", of course, because the histones come in four flavors, not just one or two, and because the H2a/H2b heterodimers "cap" the ramp and (apparently) prevent its propagation. The CENP-A/H4 "di-tetrasomes" of this paper, the H3/H4 "octasomes" of Nozama et al, the archaeal "slinkies" of Mattioli et al, and the helical "stack" of Fig. S5D all show that in the absence of a defined terminator (presumably H2a/H2b, you can indeed get extended helical ramps, either (when reconstituted with short DNA) of two heterotetramers or (when reconstituted with long DNA) of many. But those findings do not by themselves resolve the problem that you can't dock an ordinary CENP-A nucleosome into the observed cryo-EM structures of metazoan CCAN complexes. That is, it would be just as hard to jam the structure determined here into the metazoan CECAN and preserve the CENP-N interaction seen in the crystal structure with a nucleosome (indeed, probably harder) as it would to jam a CENP-A nucleosome into it. The structures determined here are therefore sound and interesting, but do not solve the problem the authors pose in the title.

This reviewer suggests that the authors recast the paper as a simple report of the new structures (Figs. 1 and 2 and S5D) and eliminate the computational molecular dynamics (which is largely hand-waving; moreover, on the experimental side, relative instability does not imply functional dynamics). The authors are then entitled to a paragraph or two of speculation about potential roles in kinetochore assembly, etc. Too much speculation will just add confusion to an already muddled issue. The authors are correct to point out that there may be different modes of association at different stages, but detailed pictures (as in their last figure) are unwarranted, as they dupe the casual reader into thinking that there is specific evidence for that model, rather than the possibility of different interactions at different steps.

In reorganizing the MS, the authors ought also to condense their unnecessarily verbose writing. The figures I designate above speak for themselves. Phrases such as "starting our analysis from the dyad" just distract. The analysis is trivial. If you start to wrap DNA around a tetrasome and then add a second tetrasome twofold related to the first, of course the new global dyad will be between tetrasomes, not within them, and the internal dyads (relating CENP-A within a tetramer) will be local. If the authors don't understand symmetry, then they should learn about it. Likewise phrases such as "we next" just distract. The reader cares about the results, not what the authors pretend was done "next".

Referee #2 Review

Report for Author:

The centromere specific histone H3 variant CENP-A a key epigenetic determinant of centromere identity and acts as structural platform, on which the centromere-associated network of proteins (CCAN) form to assemble the kinetochore. It has been generally accepted that CENP-A can form the octasomic nucleosome, containing 2 copies each of CENP-A, H4, H2A and H2B. However, several structural variants of CENP-A-containing nucleosome-like structures have been proposed, including hemisome. In the manuscript, Ali-Ahmad et al. report that CENP-A can form another type of octasome, 2x(CENP-A/H4)-di-tetrasome, which consists of 4 copies each of CENP-A and H4, and resolved its ~4 Å cryo-EM structure. Key conclusions are: (1) Two (CENP-A/H4)₂ tetramers assemble into a 2x(CENP-A/H4)₂ di-tetrasome on DNA in vitro, similar to the assembly of two (H3/H4)₂ tetramers (or previously reported H3-H4 octasome). (2) The di-tetrasome adopts an architecture distinctly different from the canonical nucleosomes, packing longer DNA (342 bp) and extended height. (4) CENP-N/L, components of the CCAN, preferentially binds to the (CENP-A/H4)₂ over the di-tetrasome, whereas CENP-C binds to both.

Overall, evidence to support the major conclusion that CENP-A can form di-tetrasome in vitro is compelling. This is

largely consistent with the essentially identical conclusion by the very recent publication by Kawasaki et al (PMID 40129080), which reported 3.66 Å resolution structure of "CENP-A-H4 octasome - equivalent to 2x(CENP-A/H4)-di-tetrasome, the term used by Ali-Ahmad et al. Neither of these two papers validated the existence of 2x(CENP-A/H4)-di-tetrasome / CENP-A-H4 octasome in cells. The major original data in this manuscript is the investigation of how 2x(CENP-A/H4)₂ di-tetrasome and (CENP-A/H4)₂ tetrasome can interact with the CCAN components, CENP-C and CENP-N/L. However, these interaction analyses were solely based on EMSA with questionable quality. Furthermore, since the authors showed that CENP-A/H4 tetrasome can recruit CENP-N/L, the proposed function of 2x(CENP-A/H4)₂ di-tetrasome is ambiguous.

Major comments:

1. For the protein samples used in the cryo-EM analysis, the authors should assess the quality of the di-tetrasome preparation using methods such as mass photometry and Western blotting. As shown in Figure S1A-B, the sample appears to be a mixture of tetrasomes and di-tetrasomes. In Figure S2B, the authors present 3D classes with more open structures.
2. Binding of CENP-C, CENP-N, and CENP-N/L to 2x(CENP-A/H4)₂ di-tetrasome or other nucleosome variants was assessed solely by EMSA (Fig. 4 and Fig. S7). Although these data are inspiring, the overall quality of the EMSA is not good enough to make quantitative assessment to support the authors' conclusions (see below). This is in part due to the heterogeneous nature of the input materials, especially 2x(CENP-A/H4)₂ di-tetrasome. As noted above, if the cryo-EM for 2x(CENP-A/H4)₂ di-tetrasome was done with the sample that contains at least two complexes that showed distinct migration patterns, how confidently you know that the upper band and the lower band represents the (CENP-A/H4)₂ tetrasome and 2x(CENP-A/H4)₂ di-tetrasome, respectively? If the upper band represents the (CENP-A/H4)₂ tetrasome as the authors claim (Fig. S1A), why are the kinetics of CENP-N/L binding to the tetrasome so different between the left and the right panels? To confirm the protein composition of each band, western blotting can be done. It would be also important to test the binding with alternative methods, preferably, using purified 2x(CENP-A/H4)₂ di-tetrasome.
3. Line 260, the authors state, "(CENP-A/H4)₂ tetrasomes but not di-tetrasomes bind CENP-N/L complex. However, the descriptions of the Fig. 4B contradicts with this conclusion, suggesting to me that the di-tetrasomes do show some binding to CENP-N/L. As noted above, if the upper band shown in the input represents the tetrasome, binding affinity of the tetrasome and the di-tetrasome may be comparable. It would be important to directly compare binding of CENP-N/L to CENP-A nucleosome and CENP-A/H4 tetrasome, if the authors argue that the tetrasome plays a critical role in CENP-N/L recruitment as depicted in Fig. 5.
4. The essentially same structure, the CENP-A/H4 "octasome", was recently reported by Kawasaki, Nozawa and colleagues (PMID: 40129080) at 3.66 Å resolution. The authors must cite this paper and discuss. Kawasaki suggests that the RG-loop of the CENP-A/H4 octasome is accessible to CENP-N, while Ali-Ahmad suggests that CENP-N/L preferably binds CENP-A/H4 tetrasome. This post should be clarified.
5. In Fig. 5, I would expect to see the major conclusion related to the 2x(CENP-A/H4)₂ di-tetrasome. However, no 2x(CENP-A/H4)₂ di-tetrasome is shown in the figure and in figure legends, questioning what the proposed function of the 2x(CENP-A/H4)₂ di-tetrasome is.

Minor comments:

1. Since The term H3-H4 octasome was published in 2022 by Nozawa et al, introduction of an alternative term "2x(CENP-A/H4)₂ di-tetrasome" does not seem justifiable. Unless the authors argue the difference between their structure and the recently published structure of CENP-A-H4 octasome, I encourage the authors use the term CENP-A-H4 octasome to avoid unnecessary confusions.
2. Please show the quality assessment of the purified proteins used in this study
3. In Figure S1C, the authors used bioanalyzer to show the protection of nucleosomes by MNase treatment. Since the data analyzed DNA lengths after extraction from chromatin, this analysis is justifiable only if the input material has comparable quality of the nucleosome or its variant. According to Fig. S1A, a large fraction of 2x(CENP-A/H4)₂ di-tetrasome sample is free DNA, whereas free DNA represents minor population in other samples. Therefore, apparent MNase sensitivity in Di-tetrasome is likely to be caused by the existence of the free DNA. Native gel analysis is preferable but I understand that this is also challenging unless the authors managed to purify the di-tetrasome.
4. In the cryo-EM grid preparation, didn't you quench the glutaraldehyde crosslink? Please clarify this.
5. In Figure S7B, at the left, once CENP-N1-240 added, 2x(CENP-A/H4)₂ started showing distinct and clear lower bands. What are those lower shifted bands?
6. In Figure 3B, the yellow highlights are missing in left side of 2x(CENP-A/H4)₂
7. Line 325-326, this statement (especially "all other histones") is misleading. For example, H3/H4 tetramers are stable.

8. Line 27, typo "tetrosomes" -> "tetrasomes"
9. Line 159, typo "then" -> "than"
10. Line 482, typo "2x(CEN-P/H4)2" -> "2x(CENP-A/H4)2"

Referee #3 Review

Report for Author:

CENP-A nucleosomes direct the formation of the centromere through the interaction with the CCAN complex. Understanding the form of chromatin bound CENP-A in complex with the CCAN is an important to determining how centromeres function and are epigenetically maintained. The manuscript attempts to addresses a conundrum in the field, that while CENP-N directly binds the CENP-A nucleosome in isolation, in the recently solved Cryo-EM structure of the entire CCAN, CENP-N does not show a direct interaction with CENP-A.

The manuscript provides a detailed description of the CENP-A tetrasome and di-tetrasome that includes Cryo-EM analysis, and molecular dynamics. These analyses identified unique features of the CENP-A tetrasomes relative to CENP-A the nucleosome and show the similarities to the H3:H4 tetrasome. This analysis is well done and exhaustive. Line 222 refers to the low affinity of binding to of CENP-C to the CENP-A tetrasome and di-tetrasome, but I do not see where the values are calculated and compared in the manuscript. It is unclear what is happening with binding of CENP-C-CR in figure 4A. CENP-C binding does not appear to be creating a consistent structure based on the loss of banding (smear) in the gel.

My primary concern with the manuscript is that the in vivo relevance of the CENP-A tetrasome and di-tetrasome is unknown. Moreover, no data are provided to suggest these complexes impact the recruitment of CENP-N/L. Although the complexes that are formed may differ based on the gel shifts in vitro, both the tetrasomes and nucleosomes bind to CENP-N. The CENP-NL complex binding to the nucleosome was not directly tested in these experiments, so it is unclear if there is a difference from the tetrasome. The experiments do not provide new insight into CENP-N/L binding to subforms of the CENP-A nucleosome.

May 7, 2025

Re: Life Science Alliance manuscript #LSA-2025-03377-T

Dr. Nikolina Sekulic
University of Oslo
Centre for Molecular Medicine Norway
Gaustadalleen 21
Oslo 349
Norway

Dear Dr. Sekulic,

Thank you for transferring your manuscript entitled "Non-nucleosomal (CENP-A/H4)₂ - DNA complexes as a possible platform for centromere organization" to Life Science Alliance. As noted in our offer to transfer, we invite you to submit a revised manuscript that addresses these points raised in during the review process another journal:

- Points 1 and 2 from Reviewer 2 to report the purity of the di-tetrasome preparation and the composition of the EMSA bands in Fig 4.
- Point 4 from Reviewer 2 to cite and discuss the recent work by Kawasaki, Nozawa and colleagues.

The revised manuscript will be returned to Reviewer 2 for evaluation. Please let me know if you have any questions about this process.

Thank you for this interesting contribution to Life Science Alliance. We are looking forward to receiving your revised manuscript.

Sincerely,

B. MANUSCRIPT ORGANIZATION AND FORMATTING:

Point-by-point response to reviewer 2:**Referee #2:**

The centromere specific histone H3 variant CENP-A a key epigenetic determinant of centromere identity and acts as structural platform, on which the centromere-associated network of proteins (CCAN) from to assemble the kinetochore. It has been generally accepted that CENP-A can form the octasomic nucleosome, containing 2 copies each of CENP-A, H4, H2A and H2B. However, several structural variants of CENP-A-containing nucleosome-like structures have been proposed, including hemisome. In the manuscript, Ali-Ahmad et al. report that CENP-A can form another type of octasome, 2x(CENP-A/H4)-di-tetrasome, which consists of 4 copies each of CENP-A and H4, and resolved its ~4 Å cryo-EM structure. Key conclusions are: (1) Two (CENP-A/H4)₂ tetramers assemble into a 2x(CENP-A/H4)₂ di-tetrasome on DNA in vitro, similar to the assembly of two (H3/H4)₂ tetramers (or previously reported H3-H4 octasome). (2) The di-tetrasome adopts an architecture distinctly different from the canonical nucleosomes, packing longer DNA (342 bp) and extended height. (4) CENP-N/L, components of the CCAN, preferentially binds to the (CENP-A/H4)₂ over the di-tetrasome, whereas CENP-C binds to both. Overall, evidence to support the major conclusion that CENP-A can form di-tetrasome in vitro is compelling. This is largely consistent with the essentially identical conclusion by the very recent publication by Kawasaki et al (PMID 40129080), which reported 3.66 Å resolution structure of "CENP-A-H4 octasome - equivalent to 2x(CENP-A/H4)-di- tetrasome, the term used by Ali-Ahmad et al. Neither of these two papers validated the existence of 2x(CENP-A/H4)-di-tetrasome / CENP-A-H4 octasome in cells. The major original data in this manuscript is the investigation of how 2x(CENP-A/H4)₂ di-tetrasome and (CENP-A/H4)₂ tetrasome can interact with the CCAN components, CENP-C and CENP-N/L. However, these interaction analyses were solely based on EMSA with questionable quality. Furthermore, since the authors showed that CENP-A/H4 tetrasome can recruit CENP-N/L, the proposed function of 2x(CENP-A/H4)₂ di-tetrasome is ambiguous.

Major comments:

1. For the protein samples used in the cryo-EM analysis, the authors should assess the quality of the di-tetrasome preparation using methods such as mass photometry and Western blotting. As shown in Figure S1A-B, the sample appears to be a mixture of tetrasomes and di-tetrasomes. In Figure S2B, the authors present 3D classes with more open structures.

We thank the reviewer for the insightful comment. We agree that assessing the homogeneity of the di-tetrasome preparation is important. As noted, our native PAGE analysis (Figure S1A-B) already indicates the presence of two distinct species—tetrasomes and octasomes—based on their differential migration. The faster-migrating band corresponds to the octasome, which we further characterized by cryo-EM and MNase digestion.

To further confirm that the upper band on our native gel is tetrasome and the lower band is octasome, we added increasing amounts of (CENP-A/H4) to DNA and followed by EMSA how the ratio of higher and lower band is changing. When there is 2x more (CENP-A/H4) to DNA, a faster-migrating lower band becomes enriched (which is especially

noticeable in Coomassie stain) and is consistent with an octasome formation (two tetramers on one DNA), while the upper, slower band disappears, consistent with the tetrasome (main text line 87-88 and Fig. S1C). This mobility assignment is also in line with prior native-gel/MS-based characterization of H3/H4 and CENP-A/H4 assemblies reported recently in PMIDs: **40129080** (Kawasaki et al., 2025) and **36322721** (Nozawa et al., 2022).

Furthermore, as reviewer suggested, we used mass photometry, which revealed three main populations: ~71 kDa (free DNA), ~134 kDa (CENP-A/H4 tetrasome), and ~187 kDa (CENP-A/H4 octasome) (Response Fig.1) that are also detected in our native gels. However, we believe that here the relative ratio of species is shifted towards more stable species due to the dilution conditions required for mass photometry (20 nM). As described in the manuscript, octasomes are dynamic and less compact than canonical nucleosomes, and their assembly equilibrium may shift toward tetrasomes at low concentrations.

We note that our cryoEM maps contain only CENP-A and H4, that are confirmed by SDS-PAGE (Fig. S1D) and no additional histones or CCAN components were added to this

sample. While Western blotting could confirm the presence and integrity of these histones, it would not distinguish between tetrasomes and octasomes.

Together, these biochemical controls establish band identities and document that the preparation contains both species (tetrasomes and octasomes), as expected from the assembly equilibrium that depends on tetramer:DNA ratio.

We appreciate the reviewer's observation on more open classes in Figure S2D. The open 3D classes in Figure S2B reflect the inherent flexibility of the octasome particles, which we discuss in the manuscript. Importantly, the 3D classes shown represent the most defined and predominant conformations captured during cryo-EM processing. We note that additional conformations likely exist but are less well-resolved due to particle heterogeneity. Furthermore, we acknowledge that interactions at the air-water interface (AWI) during grid preparation may contribute to partial DNA unwrapping or conformational bias, especially for dynamic particles like octasomes. These AWI-related artifacts and their effects on particle integrity and class distributions are well documented in the field. Relevant PMIDs: **30932812** (D'Imprima et al., eLife 2019), **30250056** (Noble et al., Nat Methods 2018), **29867291** (Glaeser, Curr Opin Colloid Interface Sci 2018), and **30804214** (Lyumkis, J Biol Chem 2019).

2. Binding of CENP-C, CENP-N, and CENP-N/L to $2x(\text{CENP-A/H4})_2$ di-tetrasome or other nucleosome variants was assessed solely by EMSA (Fig. 4 and Fig. S7). Although these data are inspiring, the overall quality of the EMSA is not good enough to make quantitative assessment to support the authors' conclusions (see below). This is in part due to the heterogeneous nature of the input materials, especially $2x(\text{CENP-A/H4})_2$ di-tetrasome. As noted above, if the cryo-EM for $2x(\text{CENP-A/H4})_2$ di-tetrasome was done with the sample that contains at least two complexes that showed distinct migration patterns, how confidently you know that the upper band and the lower band represents the $(\text{CENP-A/H4})_2$ tetrasome and $2x(\text{CENP-A/H4})_2$ di-tetrasome, respectively? If the upper band represents the $(\text{CENP-A/H4})_2$ tetrasome as the authors claim (Fig. S1A), why are the kinetics of CENP-N/L binding to the tetrasome so different between the left and the right panels? To confirm the protein composition of each band, western blotting can be done. It would be also important to test the binding with alternative methods, preferably, using purified $2x(\text{CENP-A/H4})_2$ di-tetrasome.

Because EMSA band shifts do not reliably yield accurate rate or affinity parameters, we limit our interpretation to qualitative trends. Band identities of the input species were established by a DNA-titration EMSA (upper/slower = tetrasome; lower/faster = octasome) as it was explained in previous response (**Fig. S1C-D**).

To directly address band assignment in the binding assays, we performed an additional side-by-side EMSA in which CENP-N/L was titrated against free DNA and against pre-formed tetrasome under identical conditions (**Response Fig 2**). This experiment shows that the sharp band observed upon CENP-N/L binding corresponds to the **DNA/CENP-N/L complex**, and not as expected to be tetrasome/CENP-N/L. In contrast, tetrasome/CENP-N/L, octasome/CENP-N/L, and nucleosome/CENP-N/L all present as broadened/smear signals, indicating heterogeneous and flexible engagement. In

agreement with this finding, we have adjusted the text and interpretation of our results (main text, line 260 and lines 262-267).

3. Line 260, the authors state, "(CENP-A/H4)₂ tetrasomes but not di-tetrasomes bind CENP-N/L complex. However, the descriptions of the Fig. 4B contradicts with this conclusion, suggesting to me that the di-tetrasomes do show some binding to CENP-N/L. As noted above, if the upper band shown in the input represents the tetrasome, binding affinity of the tetrasome and the di-tetrasome may be comparable. It would be important to directly compare binding of CENP-N/L to CENP-A nucleosome and CENP-A/H4 tetrasome, if the authors argue that the tetrasome plays a critical role in CENP-N/L recruitment as depicted in Fig. 5.

We have corrected the text and legends to state that CENP-N/L binds all CENP-A-containing assemblies under our conditions. As shown in Response Fig. 3, a matched titration of CENP-N/L with free DNA, CENP-A/H4 tetrasome, (CENP-A/H4)₂ octasome, and CENP-A nucleosome demonstrates that at ~5× CENP-N/L the bands corresponding to tetrasome, octasome, and nucleosome are each depleted and migrate into higher-molecular-weight complexes, indicating engagement with CENP-N/L.

4. The essentially same structure, the CENP-A/H4 "octasome", was recently reported by Kawasaki, Nozawa and colleagues (PMID: 40129080) at 3.66 Å resolution. The authors must cite this paper and discuss. Kawasaki suggests that the RG-loop of the CENP-A/H4 octasome is accessible to CENP-N, while Ali-Ahmad suggests that CENP-N/L preferably binds CENP-A/H4 tetrasome. This post should be clarified.

We now cite and briefly discuss the recently published CENP-A/H4 "octasome" structure from Kawasaki and Nozawa (PMID: 40129080) (main text: lines 90-91, lines 101-102, lines 289-290 and lines 299-303) and note that our architectural interpretation is in line with their model. Building on that framework, our added binding experiments indicate that CENP-N/L engages all CENP-A-containing assemblies (tetrasome, octasome, and nucleosome). We reconcile this with a placement model in which the octasome/nucleosome's second DNA wrap can sterically interfere with CENP-L unless conformational rearrangement or partial unwrapping occurs, making octasome/nucleosome engagement condition-sensitive; by contrast, the tetrasome lacks the second wrap and presents more accessible RG loops, which may facilitate initial contact even if a discrete band is not observed under our conditions (main text lines 297-299 and **New Supp Figure S8**).

5. In Fig. 5, I would expect to see the major conclusion related to the 2x(CENP-A/H4)₂ di-tetrasome. However, no 2x(CENP-A/H4)₂ di-tetrasome is shown in the figure and in figure legends, questioning what the proposed function of the 2x(CENP-A/H4)₂ di-tetrasome is.

Thank you for this suggestion. We depict the tetrasome in Fig. 5 (rather than the octasome) for two connected reasons. First, CENP-N/L localization depends on, and directly engages, the CENP-A RG loop, with support from both in vivo and in vitro studies

(Weir et al., 2016; Pentakota et al., 2017; Chittori et al., 2018). In parallel, CCAN-nucleosome structures emphasize linker-DNA binding, and the contribution of the RG loop within that assembled complex remains unresolved (Yatskevich et al., 2022; Pesenti et al., 2022). Second, based on our structure and EMSA data—and consistent with the recently reported CENP-A/H4 octasome architecture that includes a second DNA wrap (Kawasaki et al., 2025)—we propose that the tetrasome can act as an intermediate that presents the RG loop without a second wrap, offering a plausible configuration for full CCAN engagement via the RG loop.

Given the equilibrium between octasome and tetrasome and dynamic sliding/rearrangement of the tetramer on DNA, we do not hard-separate these species in the schematic; instead, we note that octasome→tetrasome rearrangement is likely important for—or potentially induced by—CCAN binding. The legend now states explicitly that the two illustrated routes—CCAN bound to linker DNA and CCAN bound to the CENP-A RG loop of a tetrasome—are examples, not the only configurations consistent with our data and prior literature (Weir et al., 2016; Pentakota et al., 2017; Chittori et al., 2018; Yatskevich et al., 2022; Pesenti et al., 2022; Kawasaki et al., 2025).

To soften our working hypothesis and avoid implying a single mechanism, we moved former Fig. 5 to the Supplement (now Supp Fig. S9). In its place, we present an AlphaFold-Multimer model that offers a plausible alternative in which CCAN engages the CENP-A RG loop (rather than linker DNA), shown in the new Figure 5 as a possible—not definitive—binding mode.

Minor comments:

1. Since The term H3-H4 octasome was published in 2022 by Nozawa et al, introduction of an alternative term "2x(CENP-A/H4)₂ di-tetrasome" does not seem justifiable. Unless the authors argue the difference between their structure and the recently published structure of CENP-A-H4 octasome, I encourage the authors use the term CENP-A-H4 octasome to avoid unnecessary confusions.

We have followed the reviewer suggestion and now changed the di-tetrasome term to octasome in all the manuscript.

2. Please show the quality assessment of the purified proteins used in this study
SDS-Page gels are now added to figure S1C

3. In Figure S1C, the authors used bioanalyzer to show the protection of nucleosomes by MNase treatment. Since the data analyzed DNA lengths after extraction from chromatin, this analysis is justifiable only if the input material has comparable quality of the nucleosome or its variant. According to Fig. S1A, a large fraction of 2x(CENP-A/H4)₂ di-tetrasome sample is free DNA, whereas free DNA represents minor population in other samples. Therefore, apparent MNase sensitivity in Di-tetrasome is likely to be caused by the existence of the free DNA. Native gel analysis is preferable but I understand that this is also challenging unless the authors managed to purify the di-tetrasome.

We agree that the MNase/bioanalyzer trace in Fig. S1C is suitable for quantitative footprinting when quality and homogeneity of the input is comparable. Our aim in this experiment was qualitative: to assess whether the (CENP-A/H4)₂ assembly infers

detectable protection on a 145-bp DNA and whether the protected length is shorter than for a canonical nucleosome, rather than following the kinetics of band disappearance. CENP-A nucleosomes typically protect ~120 bp (vs. ~145–147 bp for H3), and subnucleosomal CENP-A assemblies are expected—given their flexibility and reduced wrap—to protect shorter segments (~100 bp) under native conditions. Consistent with this, our CENP-A tetrasome/octasome lanes show greater MNase susceptibility and a shift toward shorter fragments compared with H3-containing complexes. We deliberately refrain from interpreting band intensities (which depend on input amount and recovery); our interpretation relies on the fragment-length distribution and the appearance of smaller products. Note also that under our digestion conditions free DNA is rapidly degraded to very short fragments that are not retained during purification preceding the bioanalyzer injection, and thus does not account for the protected-length species observed (see Fig. S1C and the gel below).

Mnase digetion of free DNA with reactions quenched at different time points 0, 2, 5 and 10 minutes.

4. In the cryo-EM grid preparation, didn't you quench the glutaraldehyde crosslink?

No, the crosslinking was done on ice for 40 minutes and the sample was immediately applied on the grid and plunge frozen. This is now stated in the Material and Methods (main text, lines 440-441).

5. In Figure S7B, at the left, once CENP-N1-240 added, 2x(CENP-A/H4)2 started showing distinct and clear lower bands. What are those lower shifted bands?

As noted in Fig S7B the Binding on CENP-N to both DNA and octasome create a ladder like profile on the gel, and this has been observed in several previous publications (eg, Allu et al., 2019). The lower band on the gel probably corresponds to DNA/CENP-N complex.

6. In Figure 3B, the yellow highlights are missing in left side of 2x(CENP-A/H4)2

Thanks for pointing this out, we added the yellow highlights in both orientations.

7. Line 325-326, this statement (especially "all other histones") is misleading. For example, H3/H4 tetramers are stable.

The original whole original paragraph and discussion was changed and softened (main text, lines 288-323):

It has been proposed that H3.3 is loaded into centromeric chromatin in S phase as a "placeholder" and then replaced by CENP-A at the exit of mitosis (49), but it is not clear how H3.3 is directed to specific sites in centromeric chromatin and whether the replacement in G1 involves the entire nucleosome or only the subnucleosomal complexes. Interestingly, Bodor et al. (50) found in a follow-up study that CENP-A is loaded in early G1 together with H4, but without concomitant loading of H2A/H2B. Moreover, the CENP-A/H4 subnucleosomal core remains stably incorporated over multiple cell divisions, whereas all other histones turn over quite rapidly. The exceptional chromatin stability of the CENP-A/H4 subnucleosomal core is directed by the CENP-A Targeting Domain (CATD) (51). The CATD comprises the L1 loop (containing CENP-ARG-loop) and the $\alpha 2$ helix of CENP-A and is required for the binding of HJURP and targeting of CENP-A to the centromere (51, 52). However, the chromatin stability of CENP-A/H4 observed by Bodor et al. (50) was not found to be dependent on HJURP. This result could be explained in part with strong hydrophobic association between H4 and CENP-A (25, 52) and with our model where, immediately after chromatin incorporation, (CENP-A/H4)₂ tetrasomes bind CCAN via CENP-N and through the CENP-ARG-loop.

Has been changed to

It has been proposed that H3.3 is deposited at centromeres during S phase as a "placeholder," later replaced by CENP-A at mitotic exit (49). How H3.3 is targeted within centromeric chromatin, and whether the G1 replacement entails exchange of a whole nucleosome or subnucleosomal complexes, remains unsettled. Consistent with this latter possibility, Bodor et al. (50) reported that CENP-A loads with H4 in early G1 in the absence of H2A/H2B, and that the resulting CENP-A/H4 subnucleosomal core shows unusual persistence at centromeres across multiple cell cycles. The CENP-A Targeting Domain (CATD)—comprising the L1 loop (including the RG loop) and $\alpha 2$ helix—is required for HJURP binding and centromere targeting (51, 52), yet the long-term chromatin stability observed by Bodor et al. (50) does not depend on HJURP. This stability is plausibly supported by the strong hydrophobic CENP-A–H4 interface (25, 52) and, in our model, by early CCAN engagement in which newly incorporated (CENP-A/H4)₂ tetrasomes interact with CENP-N via the CENP-A RG loop, thereby reinforcing retention without invoking broader claims about other histones.

to remove the over-generalization about "all other histones," focus the statement on centromeric chromatin, and align precisely with the Bodor findings cited in our manuscript.

8. Line 27, typo "tetrsoemes" -> "tetrasomes"
This is now fixed.

9. Line 159, typo "then" -> "than"

This is now corrected

10. Line 482, typo "2x(CEN-P/H4)2" -> "2x(CENP-A/H4)2"

This is now corrected

November 20, 2025

RE: Life Science Alliance Manuscript #LSA-2025-03377-TR

Nikolina Sekulić

Dear Dr. Sekulić,

Thank you for submitting your revised manuscript entitled "Structure and Dynamics of CENP-A/H4 Octasome Reveal a possible Intermediate in Centromeric Chromatin". As we indicated previously, we returned this to the original Reviewer 2, shown below as Reviewer 1, whose comments are below. We appreciate your patience during the re-review process which was slowed down by editor availability.

As you will see Reviewer 2 is overall content with the changes made to resolve their prior concerns. Please either include an estimation of tetrasome/octasome content in the assays shown in Figure S1 or state that the relative amounts of these species was not assessed. Please also check that the DNA sequence used in this assay is already shown in the Methods section. The editors appreciate the concern of this reviewer raised in point 2, however we feel these statements, which are carefully construed, are acceptable in their current form. Please also attend to the minor points raised by this reviewer. We would be happy to publish your paper in Life Science Alliance pending these changes as well as final revisions necessary to meet our formatting guidelines.

- Please be sure that the authorship listing and order is correct.
- Please upload all figure files as individual ones, including the supplementary figure files; all figure legends should only appear in the main manuscript file.
- It is recommended to exclude figures from the manuscript text.
- Please add ORCID ID for the secondary corresponding author - they should have received instructions on how to do so.
- The titles in both the system and the manuscript file must be consistent with each other.
- Please add your main, supplementary figure, and table legends to the main manuscript text after the references section.
- Please add callouts for Figures 5A; S1D and E; S2A, C-E; S3A-C, E,F; S4B; S5A-C; S6B-D and S8A-B to your main manuscript text.
- Please add scale bars for cryo-EM micrographs in S2, S3, and S5.
- If you wish you may include Figure S9 as a Graphical Abstract. To make this change, remove this figure file and upload this image with the file type "graphical abstract." Please note graphical abstracts only appear online and are not included in the published pdf.

A. FINAL FILES:

- An editable version of the final text (.DOC or .DOCX) is needed for copyediting (no PDFs).
- High-resolution figure, supplementary figure and video files uploaded as individual files: See our detailed guidelines for preparing your production-ready images, <https://www.life-science-alliance.org/authors>
- Summary blurb (enter in submission system): A short text summarizing in a single sentence the study (max. 200 characters)

including spaces). This text is used in conjunction with the titles of papers, hence should be informative and complementary to the title. It should describe the context and significance of the findings for a general readership; it should be written in the present tense and refer to the work in the third person. Author names should not be mentioned.

B. MANUSCRIPT ORGANIZATION AND FORMATTING:

Thank you for your attention to these final processing requirements. Please revise and format the manuscript and upload materials as soon as you are able.

Sincerely,

Reviewer #1 (Comments to the Authors (Required)):

The authors have adequately responded to most of the our original criticisms, but points below must be addressed before publication.

Major points:

1. Figure S1E. We originally criticized that differential MNase sensitivities between samples cannot be compared without showing the quality of the nucleosome/octasome/tetrasome that used in this assay. The authors argued that inclusion of free DNA essentially does not affect the outcomes, but it is still important to see the relative population of octasome and tetrasome. Also, please state the DNA sequence used in this assay since the relative tetrasome/octasome population seems to be affected by DNA sequences as shown in Figure S1B. At minimum, the caveats of this assay must be described.

2. Lines 269-272. "We also attempted to visualize the (CENP-A/H4)₂/CENP-N/L complex by cryo-EM but could only obtain low-resolution density maps so far.

Finally, we asked if (CENP-A/H4)₂ tetrasomes could bind CENP-HIKM directly or upon binding CENP-N/L but the results were not conclusive."

Remove these statements unless the authors provide supporting data.

Minor points:

1. Line 96 in Page 3, Figure S1C -> Figure S1E
2. Line 106 in Page 3, Figure S1C -> Figure S1E
3. Line 159 in Page 4, tetramers -> octamers?
4. Line 181 in Page 5, refer to Figure 3A

5. Line 191 in Page 5, "However, a similar movement was not observed in nucleosomes." -> Data for nucleosomes are not shown in the manuscript
6. In Figure S7B, the order of the gel does not match the order shown in the manuscript below:
Line 256 in Page 6, "(Figure S7B, middle gel)" -> Figure S7B, right gel and vice versa (Lin 259)
7. Line 267-268 in Page 6 and in Figure 4B, please annotate the tetrasome/octasome complex with CENP-N/L. Especially, which band indicates tetrasome-CENP-N/L complex?
8. Line 364, the composition of Lysis buffer is missing
9. Figure S1D is not referred in the manuscript

December 3, 2025

RE: Life Science Alliance Manuscript #LSA-2025-03377-TRR

Prof. Nikolina Sekulić
University of Oslo
Department of Medicine
Institute for basic medical sciences
Sognsvannsveien 9
Oslo 0372
Norway

Dear Dr. Sekulić,

Thank you for submitting your Research Article entitled "Structure and Dynamics of CENP-A/H4 Octasome Reveal a possible Intermediate in Centromeric Chromatin". It is a pleasure to let you know that your manuscript is now accepted for publication in Life Science Alliance. Congratulations on this interesting work.

Your manuscript will now progress through copyediting and proofing. As you indicated to the journal office, please ensure to correct the title during the proofing stage so that it is consistent with the title in our system and omits the term "non-nucleosomal". It is journal policy that authors provide original data upon request.

DISTRIBUTION OF MATERIALS:

Again, congratulations on a very nice paper. I hope you found the review process to be constructive and are pleased with how the manuscript was handled editorially. We look forward to future exciting submissions from your lab.

Sincerely,
